# Modelling the distribution of *Mustela nivalis* and *M. putorius* in the Azores archipelago based on native and introduced ranges

Lucas Lamelas-López[1]*, Xosé Pardavila[2], Paulo A. V. Borges[1], Margarida Santos-Reis[3], Isabel R. Amorim[4], Maria J. Santos[5,6]

**1** cE3c - Centre for Ecology, Evolution and Environmental Changes / Azorean Biodiversity Group, Faculty of Agriculture and Environment, Department of Environmental Sciences and Engineering, University dos Azores, Azores, Portugal, **2** Department of Cellular Biology and Ecology, University of Santiago de Compostela, Santiago, Spain, **3** cE3c –Centre for Ecology, Evolution and Environmental Changes, Faculty of Sciences, University of Lisbon, Lisbon, Portugal, **4** cE3c –Centre for Ecology, Evolution and Environmental Changes / Azorean Biodiversity Group and University of Azores, Azores, Portugal, **5** University Research Priority Program in Global Change and Biodiversity, University of Zürich, Zürich, Switzerland, **6** Department of Geography, University of Zürich, Zürich, Switzerland

* lucas.l.lopez@uac.pt

**Data Availability Statement:** All relevant data are within the manuscript and its Supporting Information files.

## Abstract

The aims of this study were to predict the potential distribution of two introduced Mustelidae, *Mustela nivalis* and *M. putorius* in the Azores archipelago (Portugal), and evaluate the relative contribution of environmental factors from native and introduced ranges to predict species distribution ranges in oceanic islands. We developed two sets of Species Distribution Models using MaxEnt and distribution data from the native and introduced ranges of the species to project their potential distribution in the archipelago. We found differences in the predicted distributions for the models based on introduced and on native occurrences for both species, with different most important variables being selected. Climatic variables were most important for the introduced range models, while other groups of variables (i.e., human-disturbance) were included in the native-based models. Most of the islands of the Azorean archipelago were predicted to have suitable habitat for both species, even when not yet occupied. Our results showed that predicting the invaded range based on introduced range environmental conditions predicted a narrower range. These results highlight the difficulty to transfer models from native to introduced ranges across taxonomically related species, making it difficult to predict future invasions and range expansion.

## Introduction

The deliberate or accidental introduction of non-native invasive species beyond their native range has been a consequence of human exploration and colonization [1]. The number of species introductions has largely increased in the last 200 years [2], and particularly in recent years due to global trade, transport and tourism [3]. Mammals were among the first organisms to be introduced by humans, either to provide food and transportation (e.g., livestock),

**Funding:** LLL (SFRH/BD/115022/2016) and PAVB and MSR (UID/BIA/00329/2019) and IAR (SFRH/BPD/102804/2014) were supported by the Fundação para a Ciência e Tecnologia - FCT. MJS was supported by the University Research Priority Program in Global Change and Biodiversity from the University of Zürich.

**Competing interests:** The authors have declared that no competing interests exist.

company (e.g., pets), support for hunting activities [1, 4] and recently control of other invasive species [4]. The introduction and spread of non-native mammals, especially predators, has been a major cause of extinctions on oceanic islands worldwide and of significant changes in the composition and structure of ecological communities [5–9]. Invasive species especially threaten native biodiversity through predation, competition, disturbance, disease transmission and facilitation of other introduced species [6, 9, 10]. Most non-native mammal species were introduced on oceanic islands during European colonization [11] and colonial nesting sea-birds are among the most negatively impacted native biota (e.g., [12]).

The Azores is an isolated oceanic archipelago in the North Atlantic where many mammal species have been introduced accidentally or deliberately since Portuguese colonization in the 15th century [13–15]. The most widespread introduced mammal species in the Azores are rodents (house mouse—*Mus musculus*, rats—*Rattus* and *R. norvegicus*), rabbits (*Oryctolagus cuniculus*), cats (*Felis catus*), dogs (*Canis familiaris*), and livestock (cows, goats, sheep and pigs). These species occur in all (rodents and cats) or almost all (rabbits) the islands [15, 16], with known impacts on native birds [17–21]. It is known that some endemic terrestrial birds went extinct, probably associated to the arrival of humans and the introduction of non-native predators [22–24]. Two species of Mustelidae have also been introduced in the Azores during the islands' colonization: the least weasel (*Mustela nivalis*) and the ferret (*M. putorius*) and both are classified as "invasive species" [25, 26]. *M. putorius* is more widespread, occurring in seven islands, and *M. nivalis* only occurs in two islands [14, 15], but both have impacted Azorean biodiversity, particularly through seabird's predation [19, 27]. It is therefore important to understand the ecological requirements of these species on both native and introduced ranges, as information on their distribution patterns, habitat requirements and abundance might inform on their potential impact on native biodiversity.

Species distribution models (SDMs) estimate the relationship between species occurrence and the environmental variables of the occurrence sites, predicting species habitat suitability [28]. SDMs are widely used in biogeography, conservation biology and ecology [29] namely to predict the potential geographical distribution of invasive species (e.g., [30]). SDMs assume that a species range is in equilibrium with environmental conditions and that this equilibrium is achieved through maintaining (or conserving) ecological niche characteristics across space and time, i.e. niche conservatism [31]. Shea and Chesson [32] suggest that invasive species success is linked to how a species responds to the new environment and the degree of niche conservatism of an invasive species might determine which regions it can invade. If introduced species exhibit strong niche conservatism, they will only occupy regions with similar conditions to those of their native range restricting their invasion potential. If introduced species exhibit loose or lack of conservatism, then they will likely invade regions not similar to those in their native ranges and likely making them more successful invaders [30, 31].

In this study we tested the hypotheses that the more similar are the environmental conditions that predict native and invaded ranges the more restricted will be the invasion potential of a species, and that this process should be consistent in closely related species. We modeled the geographic distribution of *M. nivalis* and *M. putorius* in the Azorean introduced range based on presence data from and environmental variables that influence the distribution of both species in both native and introduced ranges. Our results allow us to gain a better understanding on how the invasion potential of closely related species is related to characteristics of the native range.

## Material and methods

### Study organisms

*Mustela nivalis* **Linnaeus, 1766.**   The least weasel, *M. nivalis* is widely distributed in the Holarctic region. Its native range comprises much of Europe, northern Asia, northern Africa

and northern North America [1, 33, 34]. The species has been introduced in many other areas including Australia, New Zealand, the Netherlands, and several islands in the Mediterranean Sea (Crete, Malta, Sicily, Sardinia, Corsica, Minorca and Mallorca, [34]) and in the Atlantic Ocean (São Miguel and Terceira in the Azores, São Tomé in the Gulf of Guinea, [1, 14, 25]). *M. nivalis* has been introduced by humans to control rodent and rabbit populations [14, 34], but its geographic origin and precise time of introduction, namely to the Azores, remain unknown [35]. *M. nivalis* uses a wide range of habitats, including mixed forests, farmlands and cultivated fields, grassy fields, meadows and hedgerows [1, 33, 34]. *M. nivalis* is a specialist predator of small mammals (especially rodents), but it is also able to alter its diet according to prey availability [33, 34]. *M. nivalis* may also consume small birds, bird's eggs, frogs, salamanders, fish, worms, beetles, carrion and lizards if food is scarce [1, 33, 34]. For example, in New Zealand mice account for a large portion of *M. nivalis* diet but native birds, invertebrates and reptiles are also consumed [36]. In the Azores, *M. nivalis* has been observed visiting Cory's shearwater (*Calonectris diomedea borealis*) nests during the reproduction period, suggesting egg and/or chick predation.

*Mustela putorius* **Linnaeus, 1758.** Ferrets are the domesticated form of the albino polecat. The two species interbreed with the western polecat (*M. putorius*), and hybrids are often indistinguishable in the wild, some authors do not consider them as two separate species (e.g., [37]) or consider ferrets a subspecies of *M. putorius*, *M. putorius furo* [25, 38, 39]. We will refer to this species throughout the text as *M. putorius*.

*M. putorius* is widespread in the western Palaearctic [40]. The native range comprises western Europe from the Mediterranean north to central Scandinavia and Finland, Great Britain (but absent from Ireland), and east to about central Kazakhstan, Russia, Romania, Hungary, Czechoslovakia, Yugoslavia, eastern China and Mongolia, South to the Himalayas [1, 40]. The introduced range of this species includes Australia, New Zealand, West Indies, Japan, several islands in Great Britain [1], Mediterranean islands (including Sicily and Sardinia, [26]), and Atlantic islands (Las Palmas in the Canary Islands, Flores, Faial, Pico, São Jorge, Terceira, São Miguel and Santa. Maria in the Azores, [15, 41]). *M. putorius* has been introduced by humans to control rabbit populations [1, 26, 40]. *M. putorius* occurs in a wide variety of habitats, namely lowland woods and riparian zones, forested and semi-forested areas near water sources, rural areas close to farms and villages, marshes and river valleys, agricultural land, forest edge and mosaic habitats [1, 26, 40]. *M. putorius* is a specialist predator feeding on small mammals (mainly rabbits), but its diet varies with food availability. *M. putorius* also preys on hares, possums, birds (occasionally domestic poultry), bird eggs, lizards, hedgehogs, frogs, carrion eels and invertebrates [1, 26, 40]. In its introduced range *M. putorius* threatens native wildlife as for example ground nesting and flightless birds in New Zealand [42] and the Scottish isles [43], and seabird populations in the Azores [27].

## Study area

The study was conducted over two areas: the introduced range in the Azorean archipelago, and the native range in the European continent (Fig 1). Both species occupy Eurasian ranges, but as the geographic origin of the Azorean populations founders is still uncertain (but see [35]), we chose Europe assuming that it represents the environmental conditions that the species can use.

The two species were introduced in the Azores during the Portuguese colonization in the 15th century [14]. At the time, the archipelago was covered by Laurissilva forests, but underwent severe modification post human settlement, mainly due to the replacement of native forest by crops and pastures for cattle, and the accidental or deliberate introduction of many

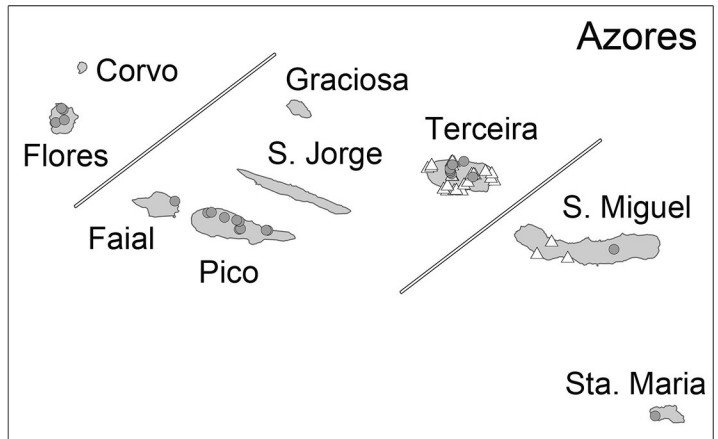
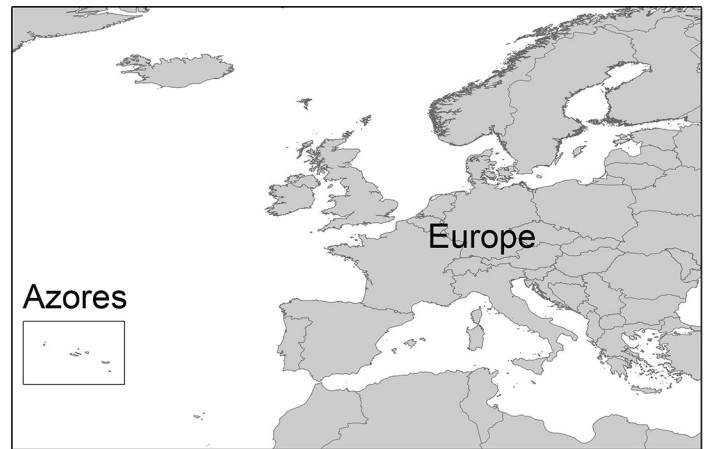

**Fig 1. Selected areas for the study of the factors affecting the distribution range of *Mustela nivalis* and *M. putorius*.** Azorean archipelago (introduced range) and Europe (native range). Occurrences data of *M. nivalis* (white triangles) and *M. putorius* (grey circles) used to perform the models are shown.

plant species, mostly for agricultural and ornamental purposes [44, 45]. The new land-uses led to the extinction of numerous endemic species, particularly in the most disturbed islands [23,

46, 47]. Currently, the landscape is relatively similar in all islands of the archipelago, with urban and rural areas being concentrated near the coast, at the lowest elevations. At intermediate elevations, the dominant land cover types include crops, pasturelands, and exotic tree plantations of the Japanese cedar (*Cryptomeria japonica*) and the Australian cheesewood (*Pittosporum undulatum*). The native vegetation remnants are only found at the highest elevations and in the most inaccessible areas [48].

## Species presence data

Species presence data were obtained from open published datasets, from standardized field sampling campaigns and from direct observations: (1) The species records in the native range were obtained from the Global Biodiversity Information Facility, an international network funded by the world's governments that provides open access data ("GBIF.org"), "iNaturalist", a citizen science platform that generate biodiversity data from species occurrences ("iNaturalist.org"), "BioDiversity4All", a citizen science platform about Portuguese biodiversity occurrence data ("biodiversity4all.org"), and "Proyecto Turón", a Spanish project that relies on naturalists and researchers participation concerning the occurrence of *M. putorius* in the Iberian Peninsula ("proyectoturon.org"). We also collected data for Spain, including direct observations (non-systematic field observations by volunteer biologists) and systematic sampling campaigns with camera-traps conducted by the authors. We only used records from the last 10 years (2007–2017) and with an accuracy higher than 1 km. For open source occurrence data, records were filtered based on the accuracy information available, i.e., records with accuracy values equal or higher than 1000 meters, as well as records with no accuracy information were removed. However, presence records from these datasets may be affected by sampling bias, given that some places are more intensely sampled than other as detections are often spatially biased towards easily accessed areas and/or records differ across the countries (S1 File) [49, 50]. This spatial bias can lead biased results from the comparison of presence records with background data drawn at random from the entire region [50]. To account for these biases we followed the approach proposed by Phillips et al. [50], and chose background data exhibiting the same bias as the presence records (for more details see S1 File). We selected a total of 3,396 and 1,616 occurrence records of *M. nivalis* and *M. putorius*, respectively, for the native range (Europe).

(2) The introduced range records were obtained from recent direct observations (non-standardized field observations by volunteer biologists, 2007—present) and standardized camera-trapping campaigns across different habitat types conducted by the authors (2013–2015), in all islands where the species occur [15]. We obtained a total of 29 and 24 occurrence records of *M. nivalis* and *M. putorius*, respectively. Currently there are no other available records in the Azores that comply with our selection criteria, i.e., reliable, recent and with an accuracy higher than 1 km.

All records were scaled to 1km$^2$ cells for both study areas (i.e., one record per 1km$^2$ cell). All occurrence records used in the study (open source and field data records) were included in S2 File.

## Environmental data

A set of candidate environmental variables was selected to test their ability to predict *M. nivalis* and *M. putorius* distributions. We selected (1) two topographic variables (e.g., slope) in order to describe the physical environment; (2) four climatic variables (e.g., annual temperature, precipitation seasonality) to describe the bioclimatic conditions; (3) a set of seven landscape level variables, including habitat type, vegetation structure (herbs, shrubs or trees), land use types,

**Table 1. Description of the candidate variables to model *M. nivalis* and *M. putorius* distribution.**

| Variable | Description | Type | Original data source |
|---|---|---|---|
| *Topographic variables* | | | |
| *altimetry | Mean altimetry | Con | WorldClim 1.4 |
| *slope | Mean slope | Con | WorldClim 1.4 |
| *Climatic variables* | | | |
| clim_bio1 | Annual mean temperature | Con | WorldClim 1.4 |
| *clim_bio7 | Temperature annual range (difference between max. and min. temperatures) | Con | WorldClim 1.4 |
| *clim_bio12 | Annual precipitation | Con | WorldClim 1.4 |
| *clim_bio15 | Precipitation seasonality | Con | WorldClim 1.4 |
| *Landscape variables* | | | |
| *landcover | Landcover map including 9 classes: (1) urban areas, (2) industrial areas, (3) agricultural areas, (4) livestock areas, (5) scrub and/or herbaceous vegetation associations, (6) forestry areas, (7) deciduous and mixed forests, (8) uncovered areas, and (9) wetlands and water bodies | Cat | CLC2006 |
| **%forest_areas | Percentage cover of coniferous, deciduous and mixed forests | Con | CLC2006 |
| %forest_coniferous | Percentage cover of coniferous forests | Con | CLC2006 |
| %forest_deciduous | Percentage cover of deciduous and mixed forests | Con | CLC2006 |
| *%scrub_herb | Percentage cover of scrub and/or herbaceous vegetation | Con | CLC2006 |
| n_patches | Number of patches | Con | CLC2006 |
| *edge_density | Total edge length of patches | Con | CLC2006 |
| *Human-disturbance variables* | | | |
| *%agricultural_areas | Percentage cover of agricultural areas (arable lands, permanent crops, pastures, heterogeneous agricultural areas) | Con | CLC2006 |
| *%artificial_areas | Percentage cover of artificial areas (urban, commercial and industrial areas) | Con | CLC2006 |
| *population_density | Density of population | Con | CIESIN 2015 |

* selected variables based on the correlation-test. Con—continuous variable, Cat—categorical variable.

and landscape heterogeneity (e.g., number of patches); and (4) three human disturbance variables, related with human population (e.g., population density) and their activities (% of artificial areas, % of agricultural areas) (Table 1). The variables were selected according to species ecology. For example, the species occur from lowlands located at sea level to alpine areas upper 2,000 m.a.s.l. and in temperate and boreal areas (i.e., different climatic and topographic conditions). The species also inhabit in a wide variety of natural (deciduous and coniferous forests, scrublands) and human-associated habitats (agricultural areas, cultivated fields, near to villages) [34, 40]. Additionally, most of these variables have been previously used in other studies about carnivore distribution or habitat use (e.g., %urban_areas, in [51]; %forest_areas, %artificial_areas, %forest_coniferous, %forest_deciduous, in [52]; clim_bio1, clim_bio7, clim_bio12, clim_bio15, altimetry, slope and aspect, in [53]). Variables are described and their source provided in Table 1. For more details see S3 File. We conducted all spatial data processing using the ArcGIS software (ArcGIS 10 ArcMap v. 10.1). All variables were scaled to 1km$^2$ grids for both study areas.

To minimize model over-fitting because of potential correlation among predictor variables, we performed a pair-wise correlation analysis to identify and exclude highly correlated variables (r > 0.75), using the Spearman rank correlation coefficient. The excluded variables were clim_bio1, %deciduous, %coniferous and n_patches. For details about this analysis see S4 File. Correlation analyses were performed with R Studio software [54].

## Species distribution modeling

Species distribution models were developed with MaxEnt v. 3.3. [55]. MaxEnt is a popular and widely used tool for species distribution modeling [56] with a predictive performance consistently among the highest performing methods [57, 58]. MaxEnt has been successfully used to produce models even from small data sets (similar to our sample size for introduced range; see Methods—Species presence data [59–61]), as those from rare or elusive species [59, 57]. These models have also been used to study invasive species, and despite the highlighted challenges inherent to modeling invaders [62], they have been shown to perform well in different phases of the invasion process (e.g., [63]). We have included a electronic supplementary material detailing the stepwise modeling process (S5 File).

## Parameter configuration

MaxEnt requires a set of parameters to be specified by the user, namely test-training percentage (i.e., the percent of presence locations to be used for model development and for internal testing), number of background points, the form of the functional relationships (feature types in MaxEnt 'language'), clamping (i.e., whether or not to constrain predictions within the range of variability of the input predictors), and regularization multiplier (i.e., to avoid over-fit of the response curves). However, there is no agreement in the literature on which set of parameter values to use in MaxEnt, and best practices suggest performing a preliminary sensitivity analysis on parameter performance for model selection (e.g., [56, 59]). Initially, we developed models to test the following parameter configurations and sets of variables: maximum number of iterations (500 and 1000), clamping (enabled, disabled), and regularization multiplier (0, 0.5, 1, 1.5, 2 and 2.5). We used 500 and 3,000 random background points for Azores and Europe models, respectively, 10 replicates, and selected the default feature type. All these parameter configurations were tested for both species (*M. nivalis* and *M. putorius*) and for both regions (Azores and Europe). We selected the best parameter configuration using the area under the curve of the receiver operating characteristic curve (hereafter AUC, [55, 56, 64]). The main advantage of this approach is that AUC provides a threshold-independent measure of model performance ([55, 56]). The AUC values vary from 0 to 1, where a value of 1 indicates perfect discrimination, a value of 0.50 indicates random predictive discrimination and values < 0.5 indicate performance worse than random [50]. However, although we used AUC to assess the parameter configuration performance and it is probably the most popular method to assess the accuracy of predictive distribution models [56] we have taken into account that exist a debate between scientists about its reliability (e.g., [65]). The parameter configuration used was: 10 runs, 500 and 3000 random background points for introduced and native ranges, respectively [66], 30% random test percentage, 1,000 maximum iterations, no clamping, auto features, and a regularization multiplier of 1.

## Model selection

Model selection was performed by combining sets of candidate predictor variables as follows: topographic + climatic + landscape + human variables (n = 12), topographic + climatic + landscape variables (n = 9), topographic + climatic + human variables (n = 8), climatic + landscape + human variables (n = 10), topographic + climatic variables (n = 5), topographic + landscape variables (n = 6), climatic + landscape variables (n = 7), climatic + human variables (n = 6), landscape + human variables (n = 7), topographic variables only (n = 2), climatic variables only (n = 3), landscape variables only (n = 4) and human variables only (n = 3); For more details see S6 File. All these model combinations were created for both species (*M. nivalis* and *M. putorius*) and for both regions (Azores and Europe). Models were then selected based on

their information content, as measured by the small-sample size corrected Akaike Information Criteria (AICc; [68]). We ranked the candidate models by their AICc, and computed the delta AICc, i.e., the difference in AICc from any given model to the model with the lowest AICc [68]. Models with ΔAICc ≤ 2 were selected. We calculated the Akaike weights to measure the weight of evidence for a given model to be the best model in each candidate model set [67, 68]. We calculated AICc using the ENMTools software (e.g., [28]).

### Environmental variables contribution to the models

We obtained the relative contribution of the environmental variables to the geographic distribution of *M. nivalis* and *M. putorius* in their introduced and native areas from MaxEnt. Given that we obtained two equally performing top-models for *M. putorius* introduced range model, we averaged these models by calculating the weighted average of relative contributions of each variable based on their Akaike weight (e.g., [68]).

### Habitat suitability for *M. nivalis* and *M. putorius* in the Azores

Species habitat suitability maps were generated by applying MaxEnt models to each cell of the Azorean archipelago map, obtaining a Habitat Suitability Index (HSI) that varied between 0.00 and 1.00 (e.g., [69]). We created a total of four suitability maps for the Azores, two maps per study species. One habitat suitability map was based on the top-models for the species introduced range and the other was based on the top-models for the species native range. Azorean distribution prediction models were projected from the Azorean islands where the species occur (introduced range) and from Europe (native range).

We used this HSI values to perform an additional evaluation of model outputs. We evaluated the outputs from the SDMs based on their fit to a subset of presence records obtained through camera-trapping surveys (see e.g., [70]). We obtained the HSI value of each cell for which we had field species records. Then, we calculated the proportion of occurrence records with HSI values higher than 0.75 (i.e., high habitat suitability), the proportion with HSI between 0.75 and 0.50 (high-medium habitat suitability), the proportion with HSI between 0.50 and 0.25 (medium-low habitat suitability), and the proportion with HSI lower than 0.25 (low habitat suitability). We performed a $Chi^2$ test with Yates correction to evaluated if the aforementioned values were significantly different than expected by chance (i.e. in relation to amount of cells available with HSI values higher than 0.75, between 0.5 and 0.75, between 0.25 and 0.5 and lower than 0.25), using R Studio software [54].

Additionally, due to the small sample size to model in the introduced area, to assess the reliability of the introduced models, we included maps of uncertainty in predictions, based on the introduced range by overlapping n-1 models (n = number of presence records). This allowed us to determine the consensus, i.e., how many times a given cell was predicted to be suitable (HSI > 0.5) for both study species (see S7 File).

## Results

### Model selection

*M. nivalis* and *M. putorius* top-models for both the native range in the Azores and the introduced range in Europe are shown in Table 2. For *M. nivalis*, the introduced range model included only climatic variables, and native range model included topographic, climatic, landscape and human variables (see Table 2). For *M. putorius* the introduced range model included climatic variables, and the native range model included topographic, climatic and human variables.

**Table 2. Results of AIC-based model selection for the suitability of *Mustela nivalis* and *M. putorius* occurrence, in the introduced range (Azores) and in the native range (Europe).**

| | ΔAICc | $w_i$ | K | −2 log (£) |
|---|---|---|---|---|
| *Mustela nivalis* | | | | |
| **Introduced area** | | | | |
| clim_bio7 + clim_bio12 + clim_bio15 | 0.00 | 0.65 | 4 | 353.80 |
| **Native area** | | | | |
| altimetry + slope + clim_bio7 + clim_bio12 + clim_bio15 + landcover + %forest_areas + %scrub_herb + edge_density + % agricultural_areas + artificial_areas + population_density | 0.00 | 0.99 | 80 | 62754.13 |
| *Mustela putorius* | | | | |
| **Introduced area** | | | | |
| clim_bio7 + clim_bio12 + clim_bio15 | 0.00 | 0.54 | 5 | 359.20 |
| **Native area** | | | | |
| altimetry + slope + clim_bio7 + clim_bio12 + clim_bio15 + %agricultural_areas + artificial_areas + population_density | 0.00 | 0.99 | 68 | 29704.40 |

Top models are included (ΔAICc ≤ 2). ΔAICc AICc difference; $w_i$ Akaike weight; K number of parameters; −2 log (£) −2 log-likelihood.

## Predicted *M. nivalis* and *M. putorius* ranges in the Azores

We found marked differences in the potential distribution ranges for both species. The predicted distribution based on the introduced model for *M. nivalis* showed higher suitability in coastal areas of some islands (Terceira, Graciosa, Pico and São Jorge) and lower suitability inland. According to this model, the oriental islands of São Miguel and Santa Maria showed very low values of HSI. Contrarily, the native-based models showed higher HSI values inland (Fig 2). The predicted distribution for *M. putorius* showed the inverse pattern in the introduced-based models, with higher suitability towards the center of the islands and lower suitability in the coastal areas (Fig 3). Oriental islands also showed low suitability for *M. putorius*. In contrast, the native-based model for *M. putorius* showed lower HSI values inland.

## Environmental correlates of island invaders

For *M. nivalis*, temperature annual range, annual precipitation and precipitation seasonality were included in introduced-based model and altimetry, slope, temperature annual range, annual precipitation, precipitation seasonality, land-use, cover of forest areas, cover of scrubs and herbaceous areas, edge density, cover of agricultural areas, cover of artificial areas and human population density were included in the native-based model (Table 2). Temperature annual range was the variable that showed a higher relative contribution to the introduced-based model (Table 3). For the native-based model the variables that showed higher relative contributions were altimetry, cover of forest areas and cover of artificial areas.

For *M. putorius*, the top-models derived from introduced and native range data also included different variables' groups. The introduced-based model only included the climatic variables (temperature annual range, annual precipitation and precipitation seasonality) while in the native-based model the altimetry, slope, temperature annual range, annual precipitation, precipitation seasonality, cover of agricultural areas, cover of artificial areas and human population density variables were included in the model. Annual precipitation was the variable with higher relative contribution to the introduced-based model. Temperature annual range and altimetry were the variables with higher relative contribution to the native-based model (Table 3).

Validation of species models based on their fit to the field data, showed that the models performed well for both species, especially for native-based models (Table 4). More than 50% of

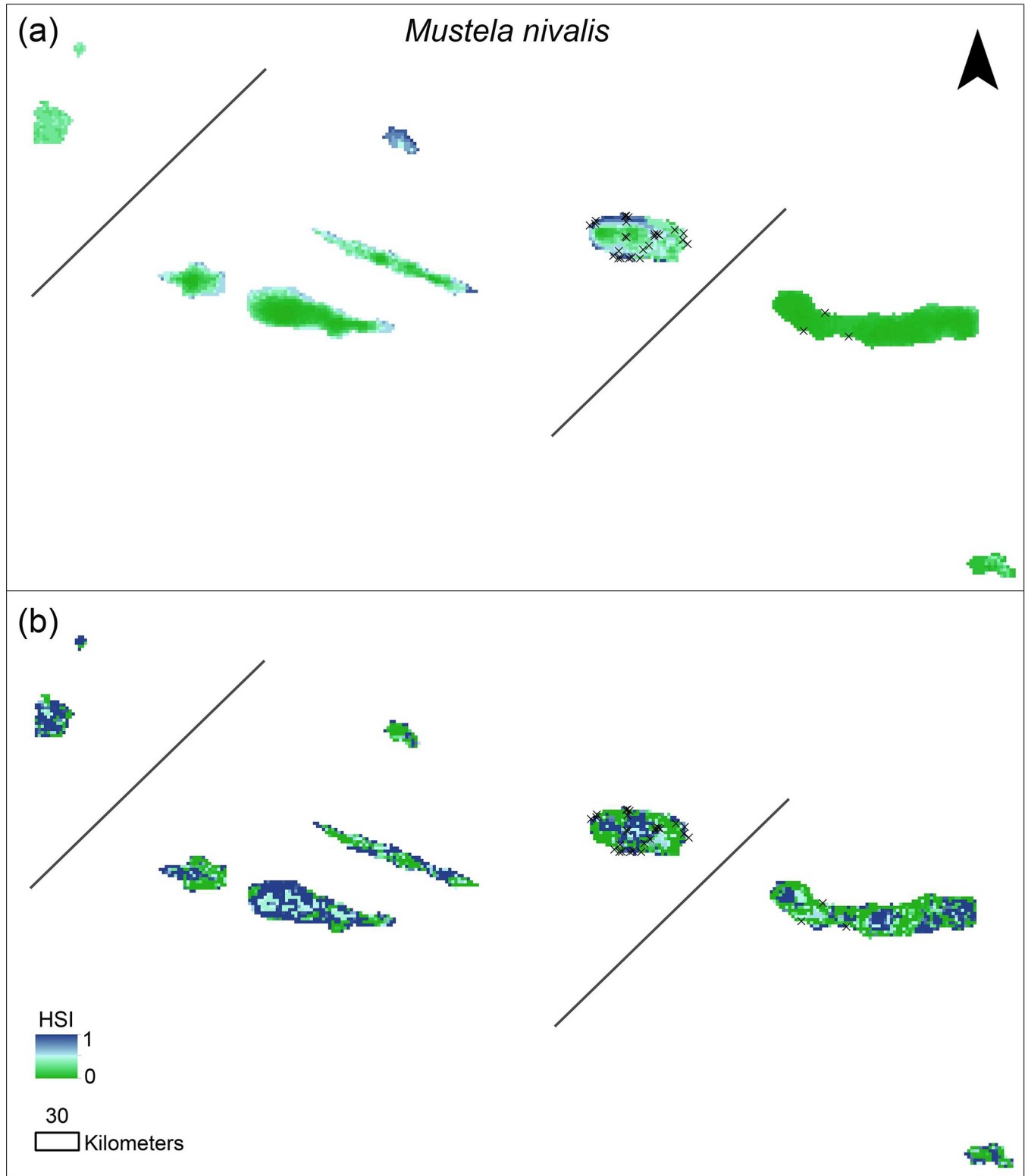

**Fig 2. *Mustela nivalis* potential distribution map for the Azores.** (a) Distribution map derived from SDM based on the introduced range; (b) distribution map derived from the SDM based on the native range. Black crosses indicate field data records. HSI, Habitat Suitability Index.

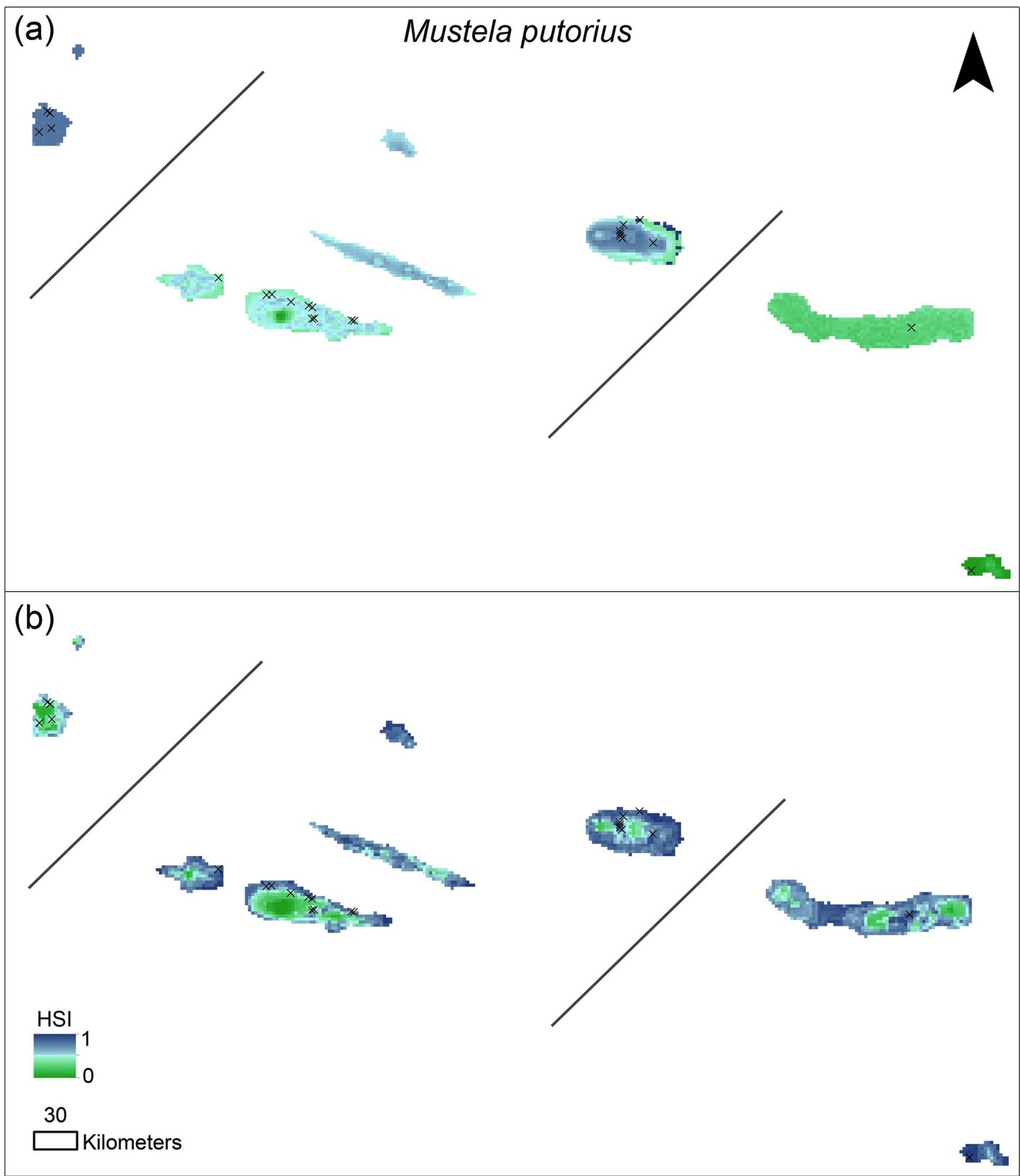

**Fig 3. *Mustela putorius* potential distribution map for the Azores.** (a) Distribution map derived from SDM based on the introduced range; (b) distribution map derived from the SDM based on the native range. Black crosses indicate field data records. HSI, Habitat Suitability Index.

**Table 3. Environmental variables contribution to the potential distribution of *Mustela nivalis* and *M. putorius* in introduced (Azores) and native (Europe) ranges.**

| | | *Mustela nivalis* | | *Mustela putorius* | |
|---|---|---|---|---|---|
| | | Introduced area | Native area | Introduced area | Native area |
| **Topographic variables** | | | | | |
| | Altimetry | — | 17.6 | — | 17.1 |
| | Slope | — | 10.6 | — | 5.6 |
| **Climatic variables** | | | | | |
| | clim_bio7 | 85.7 | 9.3 | 42.2 | 31.9 |
| | clim_bio12 | 0.0 | 3.2 | 54.3 | 11.4 |
| | clim_bio15 | 14.3 | 13.7 | 3.5 | 7.3 |
| **Landscape variables** | | | | | |
| | landcover | — | 1.1 | — | — |
| | %forest_areas | — | 16.0 | — | — |
| | %scrub_herbs | — | 2.0 | — | — |
| | edge_density | — | 10.0 | — | — |
| **Human-disturbance variables** | | | | | |
| | %agricultural_areas | — | 1.2 | — | 13.1 |
| | %artificial_areas | — | 14.9 | — | 0.2 |
| | population_density | — | 0.4 | — | 13.4 |

Values represent the weighted average of the relative contributions of each variable (%) based on top-models relative weights.

the species records occurred in cells with HSI-values higher than 0.5 for all models (with exception of *M. nivalis* introduced-range model). The $Chi^2$ test revealed significant differences in the number of records for *M. nivalis* introduced-range model between observed and random values ($Chi^2 = 44.096$, df = 3, p-value<0.01). The other models showed no significant differences between observed and random values (*M. nivalis* native-range model: $Chi^2 = 2.7308$, df = 3, p-value = 0.435; *M. putorius* introduced-range model: $Chi^2 = 5.4747$, df = 3, p-value = 0.140; *M. putorius* native-range model: $Chi^2 = 2.1504$, df = 3, p-value = 0.542). Finally, the maps of uncertainty in prediction based on the introduced range showed similar patterns that the distribution models for the introduced range (see S7 File).

**Table 4. *Mustela nivalis* and *M. putorius* species distribution model adjustment to the field data.**

| | Introduced-individuals records based SDM | Native-individuals records based SDM |
|---|---|---|
| ***Mustela nivalis*** | % records (n) | % records (n) |
| HSI > 0.75 | 14.28 (2) | 57.14 (8) |
| 0.5 < HSI < 0.75 | 28.57 (4) | 14.28 (2) |
| 0.25 < HSI < 0.5 | 35.71 (5) | 7.43 (1) |
| HSI < 0.25 | 21.43 (3) | 21.43 (3) |
| ***Mustela putorius*** | | |
| HSI > 0.75 | 9.09 (1) | 54.55 (6) |
| 0.5 < HSI < 0.75 | 63.64 (7) | 27.27 (3) |
| 0.25 < HSI < 0.5 | 27.27 (3) | 18.18 (2) |
| HSI < 0.25 | 0 (0) | 0 (0) |

The percentage (%) and number of records (n) based on field data for four categories of Habitat Suitability Index (HSI): HSI > 0.75, 0.75 > HSI > 0.50, 0.50 > HSI > 0.25 and HSI < 0.25.

## Discussion

We set out to predict for the first time the potential distribution of two introduced carnivores (*M. nivalis* and *M. putorius*) in the Azores, using species distribution models derived from native and introduced range records and environmental conditions. Wider distribution ranges based on native-based records, where species occupy most available habitats in all islands, compared to narrower ranges based on introduced-based records, suggest that both species are non-invasive in the Azores. Despite the fact that there is enough suitable habitat and that some invasive species on islands show a time lag before becoming invasive (e.g., [71]), given that both species were introduced to the Azores several centuries ago, they are not dominant and show limited distribution. The inclusion of different sets of variables in native and introduced ranges suggests that these species do not occupy habitats according to the species ecology in their native range, which in turn, according to our hypotheses, suggests that the archipelago provides potential for establishment and/or expansion, for both species.

### Introduced and native range predictions

In general, native range models showed higher HSI values, and a more widespread distribution (Figs 2B and 3B) than the introduced range models (Figs 2A and 3A), for both species, which was consistent with our second hypothesis. *M. nivalis* introduced-range models showed higher suitability in coastal areas, while the native-range models showed also high suitability at intermediate elevations. *M. putorius* introduced range models showed higher suitability at intermediate elevations, while native-range models also included high suitability in coastal areas. These differences in the prediction maps for both species should be due to the climatic conditions at the introduced area, given that only climatic variables were included in the introduced-range models. Other authors also found differences in the climatic conditions invasive species withstand in native and introduced ranges (e.g., [72, 73]). This suggests that particular conditions of the introduced environment do not mimic those at the native range, i.e., variables that influence species distribution could be novel or differ between introduced and native ranges [73]. For island invasive species, island and mainland conditions may differ even more; for example, in the Azores altimetry and slope diversity influence the climate at very fine scales [74], and thus species distribution may respond to sudden changes in topographic complexity [48, 75, 76].

Moreover, the differences in the prediction maps between both ranges could also be due to ecological processes not included in the models as, for example, biotic interactions [31, 73, 77]. Biotic interactions among species are likely to be different on the native and introduced ranges, being that in the latter natural competitors/predators of introduced species are usually absent (e.g., [6]). In our case, *M. erminea* competes with *M. nivalis* [33] and *Neovison vison* probably competes with *M. putorius* [78] in their native range, but these competitors species are absent in the Azores. Due to this absence of predators we could expect a wider distribution range and set of environmental conditions in the introduced-based models. Curiously, the introduced range models showed a more restricted distribution for both species. This could be simply because of an effect of the much smaller sample size of the introduced range, which could affect AIC values when selecting the best models. Nonetheless, model comparison was performed within study areas, so the best models should have been selected in both cases. Although our native-area model was formed by a limited number of background points (see S1 File), the native-ranges generally comprise large continental areas, and the scale could affect predictions in narrower introduced ranges, as insular systems are [79]. These differences suggest that the transferability of models from native to introduced ranges needs to be performed with caution, as it can be greatly affected by sampling sizes and might miss important

conditions only found in the introduced ranges. Further, model predictions to remote or inaccessible areas (e.g., top of mountain of Pico, located around 2,300 m.a.s.l.) should be carefully interpreted.

### *Mustela nivalis* distribution

Although *M. nivalis* uses a wide variety of habitats [1, 25, 34], it prefers rural and agricultural areas [34], and habitats that provide protection against potential predators (e.g., raptors, [33]). Consequently, *M. nivalis* potential distribution maps showed high HSI values in low and middle elevation areas, where human activities and human associated-habitats (e.g., agriculture, rural areas, etc) are concentrated. Additionally, *M. nivalis* is a specialist predator feeding on small mammals, especially small-rodents, and habitat selection is usually determined by local prey distribution [33]. Given that in the Azores rodents are more common in human-associated habitats [14, 80], rodent abundance potentially explains the *M. nivalis* distribution patterns.

   *M. nivalis* in the Azores only occurs in the most human populated islands of Terceira and São Miguel [14, 15], but the predicted potential distribution for the remaining islands showed, in general, high habitat suitability, comprising almost the entire area for the smaller islands, probably due to their extensive agricultural fields. Given that those smaller Azorean islands also hold house mice and rats, if *M. nivalis* were to be introduced, it would probably become widespread and abundant. In the larger islands (e.g., Pico island) the predicted suitability was again higher for urban and rural areas near agricultural areas. High elevation areas showed low suitability, suggesting that the native and most pristine ecosystems might remain free of *M. nivalis* or that the species might occur in low abundance. However, rodent populations also occur at higher elevations in the Azores [14, 80].

### *Mustela putorius* distribution

*M. putorius* uses different habitat types [1, 26, 40]. In the introduced areas, *M. putorius* usually occurs in grasslands, scrubs, pasture-lands, agriculture areas and urban and suburban areas [1, 26]. Therefore, our results are in line with the known *M. putorius* preferred habitats. Potential distribution maps showed higher HSI values at intermediate elevations, which are dominated by grasslands, agricultural areas, semi-natural meadows and exotic tree plantations. However, native-based models showed high HSI values also at low elevations, where urban areas and other human activities are concentrated, habitats frequently occupied by this species according to the species ecology [40]. Consequently, the native-based model included human-disturbance variables. Additionally, *M. putorius* is a predator specialized mainly in lagomorphs [81–83], and rabbits in the Azores prefer agricultural areas, grasslands and semi-natural meadows located inland at intermediate elevations. This is consistent with *M. putorius* predicted distribution with higher HSI in areas where rabbits are expected to be more abundant. *M. putorius* in the Azores occurs in most islands, except for Corvo and Graciosa [15]. The predictive maps revealed high HSI values in inland areas for *M. putorius* free islands, which suggests that an eventual introduction would possibly result in the establishment of this species on those two islands. The highest elevation areas of the Azores also showed low HSI values for *M. putorius*, which suggests that *M. putorius* is absent or in lower abundances in the most pristine native forest areas of the archipelago.

## Conclusion

SDMs are often used to predict the potential distribution of invasive species based on environmental conditions on their native range (e.g., [30]). However, factors that influence species

distribution in the introduced range could be novel or differ from those in their native ranges [73]. The difference might even be starker when the introduced ranges include islands while the native ranges are continental areas. Invasive species SDMs can be useful for the management of biological invasions but a careful interpretation is necessary and must be based on ecological knowledge [63].

In the case of the Azores, *M. nivalis* and *M. putorius* distribution patterns are mainly associated with climatic variables and human-associated habitats. We found that islands that are currently free of these species provide highly suitable habitat, being therefore important to prevent species arrival and establishment on these islands. Future studies should investigate the distribution of the two introduced carnivores based on their diet knowledge. Furthermore, given the potential impact of *M. nivalis* and *M. putorius* on native insular biodiversity, our results on the potential distribution of these introduced predators in the Azores might have important conservation implications, namely concerning seabirds' colonies. Although the real impact of these predators on seabirds in the archipelago is yet to be assessed, the few existing studies (e.g., [19, 27] suggest that weasels are potential threats to seabirds, as highly suitable areas for this predator overlap with seabird breeding areas. Modeling species invasions on islands is therefore crucial to understand invaders ecological requirements and consequences, with potential cascading effects to native fauna and ecosystems, and to decide on actionable management options.

## Supporting information

**S1 File. Selection of background points selected to model the native area conditions.** (DOCX)

**S2 File. Occurrence records of *Mustela nivalis* and *M. putorius*.** (DOCX)

**S3 File. Detailed explanation about obtained environmental data.** (DOCX)

**S4 File. Correlation analysis between variables in native and introduced ranges.** (DOCX)

**S5 File. Figure detailing the stepwise modeling process.** (DOCX)

**S6 File. Combination of sets of candidate predictor variables.** (DOCX)

**S7 File. Maps of uncertainty in predictions.** (DOCX)

## Acknowledgments

We gratefully acknowledge the following people, for support in GIS processing: Agustín Fernández and Filipe Fernandes; for providing species records and other data: Verónica Neves, Joel Bried, Luis Barcelos, Rémi Fontaine, Iván Salgado and the "Atlas de Mamíferos de Portugal" team, namely Joana Bencatel; for logistical support and field assistance: Jose Sarangollo, David Rodilla, María Olivo, Sophie Wallon and Luis Ansias; and for comments and suggestions on the manuscript: Marco Girardello, Pedro Cardoso and Artur Gil. Data on introduced range was a contribution to AZORESBIOPORTAL that supported the Open Access of this

manuscript though the project ACORES-01-0145-FEDER-000072, financed by FEDER in 85% and by Azorean Public funds by 15% through Operational Program Azores 2020.

## Author Contributions

**Conceptualization:** Lucas Lamelas-López, Margarida Santos-Reis, Isabel R. Amorim, Maria J. Santos.

**Formal analysis:** Lucas Lamelas-López.

**Investigation:** Lucas Lamelas-López, Xosé Pardavila.

**Methodology:** Lucas Lamelas-López, Xosé Pardavila.

**Resources:** Paulo A. V. Borges.

**Software:** Lucas Lamelas-López, Xosé Pardavila.

**Supervision:** Xosé Pardavila, Paulo A. V. Borges, Margarida Santos-Reis, Maria J. Santos.

**Visualization:** Lucas Lamelas-López, Xosé Pardavila.

**Writing – original draft:** Lucas Lamelas-López, Maria J. Santos.

**Writing – review & editing:** Lucas Lamelas-López, Paulo A. V. Borges, Margarida Santos-Reis, Isabel R. Amorim, Maria J. Santos.

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
