## [Decision Letter · Decision Letter 0]

14 Jan 2020

PONE-D-19-33731

Modelling the distribution of Mustela nivalis and M. putorius in the Azores archipelago based on native and introduced ranges

PLOS ONE

Dear Mr Lamelas-López,

Thank you for submitting your manuscript to PLOS ONE. After careful consideration, we feel that it has merit but does not fully meet PLOS ONE’s publication criteria as it currently stands. Therefore, we invite you to submit a revised version of the manuscript that addresses the points raised during the review process.

While reviewers agree that this is an important study, there are major concerns about study design and data analysis.

We would appreciate receiving your revised manuscript by Feb 28 2020 11:59PM. To enhance the reproducibility of your results, we recommend that if applicable you deposit your laboratory protocols in protocols.io, where a protocol can be assigned its own identifier (DOI) such that it can be cited independently in the future. For instructions see: http://journals.plos.org/plosone/s/submission-guidelines#loc-laboratory-protocols

We look forward to receiving your revised manuscript.

Kind regards,

Ulrike Gertrud Munderloh, Ph.D.

Academic Editor

PLOS ONE

Journal Requirements:

2. Our internal editors have looked over your manuscript and determined that it is within the scope of our Biodiversity Conservation Call for Papers. This collection of papers is headed by a team of Guest Editors for PLOS ONE (https://collections.plos.org/s/biodiversity). The Collection will encompass a diverse range of research articles on biodiversity conservation, including management of invasive species. Additional information can be found on our announcement page: https://collections.plos.org/s/biodiversity

If you would like your manuscript to be considered for this collection, please let us know in your cover letter and we will ensure that your paper is treated as if you were responding to this call. If you would prefer to remove your manuscript from collection consideration, please specify this in the cover letter

3.  In your Methods section, please provide additional location information of the study area, including geographic coordinates for the data set if available.

Reviewers' comments:

Reviewer's Responses to Questions

**Comments to the Author**

1. Is the manuscript technically sound, and do the data support the conclusions?

Reviewer #1: No

Reviewer #2: Partly

2. Has the statistical analysis been performed appropriately and rigorously? 

Reviewer #1: No

Reviewer #2: No

3. Have the authors made all data underlying the findings in their manuscript fully available?

Reviewer #1: Yes

Reviewer #2: No

4. Is the manuscript presented in an intelligible fashion and written in standard English?

Reviewer #1: Yes

Reviewer #2: Yes

5. Review Comments to the Author

Reviewer #1: This paper models habitat suitability in the Azores archipelago for two introduced mustelid species, Mustela nivalis and Mustela putorius. For each species, the authors fit two models: (i) one model fitted to a dataset of detections collected over the whole Europe (native range), and (ii) one model fitted to a dataset of detections collected only in the Azores islands (introduced range). For each species, the authors then compare the maps predicted by the two models (introduced and native), and show that the maps obtained with the two datasets are different. The map predicted using data collected over the whole Europe does not correspond to the map predicted using data collected in the Azores. The authors conclude that this difference indicates that species distribution models should be carefully interpreted when used to predict the range of a species in an area based on data collected in another area.

As a whole, I found this paper well written. Their aim is interesting and I fully endorse their conclusion, i.e. that "factors that influence species distribution in the introduced range could be novel or differ from those in their native ranges (...). The difference might even be starker when the introduced ranges include islands while the native ranges are continental areas. Invasive species SDMs can be useful for the management of biological invasions but a careful interpretation is necessary and must be based on ecological knowledge".

However I have concerns related to the statistical methods used in their paper. In addition, I think that this paper should contain more details on the approach used to fit the models, possibly given in supplementary data. It is presently difficult to judge of the validity of the analyses based on the information provided in the paper.

My main concerns are:

****

My first main concern is related the difference of scales of their two models (native and introduced). The concept of scale is an essential concept in ecology, and we should not expect to obtain the same results at different scales. For example, on a large scale, a savannah ungulate species might select areas close to water during the dry season, but on a small scale rarely use it rarely of the abundance of nearby predators. Thus large scale SDM might predict a high HSI for areas close to water, whereas small-scale SDM might predict low HSI for such areas. Similarly, I do not expect that differences between the distribution of a species over a whole continent with a very large diversity of climates, landscapes, human densities, etc. and the expected distribution in a set of small islands with a comparatively homogeneous climate and landscape to be automatically caused by the fact that the species is native in one place and introduced in another. This difference of scale may for example explain why "native" models predict wider range than "introduced" models. In my opinion, it would be more sensible to compare the Azores to areas with similar size and environmental characteristics (or at least as similar as possible) where the species is natively present.

****

Another important concern is the use of a very heterogeneous detection dataset to fit the models. This dataset was built from various datasources including mainly GBIF and citizen science programs. As rightly noted by the authors, such data are characterized by numerous collection bias. Thus, species detection are generally more numerous in areas where there are more observers. For example, considering the Terceira island in Fig. 2 and 3, it is clear that most detections occur in places where there are most people (Angra do Heroismo on the Eastern side, Praia da Vitoria on the southern side). The same is true for data collected in mainland Europe.

Thus, any model not accounting for these bias will be biased and predict high suitability indices in areas where there are both many observers and many animals. The only possible approach to SDM with such data implies a model of the spatial distribution of the survey effort.

The authors acknowledge this point and indicate "presence records from these datasets may be affected by sampling bias, because they are often spatially biased toward easily accessed areas. This spatial bias can lead environmental bias given that the background data are usually drawn at random from the entire region [48]. Consequently, and according to Phillips et al. [48], we chose background data exhibiting the same bias as the presence records." However, the authors do not detail how they chose such biased background data.

It seems that the authors have used the *target group* method of Phillips et al., as they cite this paper. This approach rely on the definition of a broad set of many species, chosen to represent the specimen collection or observation activities of collectors of the target species (so-called "target groups"). Is this indeed the method used here ? If this is the case, what are the species chosen for this target group ? Are the data selected from the same source?

Moreover, the Azorean data were "cleaned" by selecting one record per squared km "to avoid a potential bias of spatially auto-correlated presences". I do not see how the method of Phillips et al. could be used with such a treatment: either (i) the same "cleaning" treatment is carried out on the background data (keeping one background point per squared km) and in this case, we loose the bias correction in the model (since the greater probability to find a detection at a given place is no longer reflected by the amount of background points at this place), or (ii) the same "cleaning" treatment is not carried out on the background data, and these data "overcorrect" the bias where the species is detected (since the presence data have a "cleaning" treatment that the background data do not have, and the species can only be detected once at a given place).

I would like to see a map of the distribution of the points of other species used as background data, to have an idea of the spatial distribution of the survey effort.

This is an important aspect for a SDM, as the fitting of SDM to such a biased dataset will focus only on parts of geographic space that contain presence samples. As noted by Phillips et al. "predictions into unsampled areas, especially those with conditions outside the range observed in sampled areas, should be treated with strong caution." This is an important warning given the aim of the paper: given the small sample size of data available in the Azores island, I fear that there are areas that are not sampled at all, and this map would allow to identify which predictions should not be trusted at all.

****

Other methodological concerns are related to the small sample size of the data set used for the "introduced models":

* Note that because the "introduced" model is fitted on a very small dataset (less than 30 detections), the number of variables in the model will necessarily be much smaller than the number of variables in the "native" model based on a much larger dataset (the sample size has a very strong effect on the number of variables that the AIC will select). This difference of sample sizes makes difficult the comparison of the structure of the two separate models (in particular, it is difficult to conclude anything concerning the fact that a particular variable is present in the "native" model and not in the "introduced" model).

* Visual comparisons of maps in figure 2 and 3 are actually simply visual comparisons of point predictions by models. The prediction uncertainty is not accounted for in theses comparisons. Given the small sample size for "introduced" models, and given that some areas in the Azores islands seem to be scarcely sampled, this will probably lead to weak conclusions at these places.

*****

Detailed comments follow:

* Line 241--247. The description of the models is not clear. Would it be possible to present them in a table ? Presently, it is very difficult to understand which models have been compared. For example, the second model presented here is: "topographic + climatic + landscape variables (n=9) + topographic + climatic + human variables (n=8)". Here, "Topographic + climatic" is present twice ? I think that a more detailed explanation is required.

* Table 2: for the model "M. putorius, introduced": what is the difference between the two models since both include the same variables?

* line 282 : "For M. nivalis, one model was selected for both the native and the introduced range". Do the authors mean "one model was selected for each dataset" ? The selected model is indeed not the same for the two species.

* line 281--290. It is not necessary to include the Akaike weights both in the table and in the text.

* Table 4: there are several problems with this table, for the "introduced" models: the sum of percentage for M. putorius is >100%, and for Mustela nivalis, > 50% of the records are characterized with HSI<0.5, contrarily to what is written in the text.

Reviewer #2: GENERAL COMMENTS

This a well-written and potentially interesting study modelling two species’ distributions based on native and (insular) introduced ranges. However, some parts of the methods need further explanation or tuning, and the Discussion also needs improvements. It is also not clear where the species occurrence data collected by the authors are made publicly available.

Figure 1 shows a substantial problem of survey bias, with some countries presenting a very high and other countries a very low density of occurrence records, without this reflecting the actual occurrence patterns of the analysed species. France and Switzerland are very clear examples of countries providing wildly insufficient data in this case, but even other countries like Spain and Portugal are clearly under-represented in the analysed dataset, compared to other data sources (at broader spatial resolutions) such as the national mammal atlases, which show that these species occur in many more areas than were used in the native range models. Moreover, this bias is not related to accessible areas, and it is unclear how the Maxent bias analysis deals with it.

Other parts of the methods also need better explanation and justification, such as the limited variations in model parameters that were chosen for sensitivity analysis, as well as the limited method for model evaluation and selection. Recent specific literature on good modelling practices should be used for better tuning of model parameters.

Figures 2 and 3 show very different (sometimes nearly opposite) predictions from native-range and introduced-range models. This needs to be explored and explained more clearly – and separately – in the Discussion.

SPECIFIC COMMENTS

Ln 20: I would add “archipelago” after “Azores”, as this is a relevant biogeographic feature which may not be immediately obvious to all readers.

Ln 24-26: The current phrasing is confusing as to whether the “differences” are between species or between models. I suggest rephrasing to something like “We found differences in the predicted distributions of models based on introduced and native occurrences for both M. nivalis and M. putorius in the Azores”.

Ln 32: It its not clear what “this” refers to.

Ln 62: The Azores are more than one island, so “island’s ” should be “islands’ ”.

Ln 55-59: This paragraph refers common mammals in the Azores, but most of the provided references are focused on birds. I miss a reference to the atlas of Portuguese mammals, which seems to be mentioned in acknowledgments but is not among the cited literature.

Ln 69: “distributions” should be “distribution”.

Ln 78: Overly long sentence. I would add a comma after “conservatism”.

Ln111: Replace the final comma with a full stop and start a new sentence.

Ln 144: “deliberately” should be “deliberate”.

Ln 145: “or accidentally” is redundant in this sentence.

Ln 163: Were these direct observations and sampling campaigns conducted across the modelled range, or centered on particular regions/countries? It would also be interesting to mention how many points were added by the authors to the data available in the open published datasets, and if these data were added to these (or to which) public platforms.

Ln 164: How was this record accuracy assessed? Namely, which columns of the public databases were used for this filtering, and with which values?

Ln 166: In this case, the bias was clearly not (only) towards accessible areas, but also reflected the habits of different countries in uploading occurrence records to the analysed databases. France and Switzerland, for example, have a lot of accessible areas but a tiny portion of records.

Ln 168-169: More details are necessary on how this biased background sample was generated exactly, and a map of these background points should be included either in the article or the supplementary files.

Ln 169: I would start a new paragraph at “The introduced...”

Ln 173-174: Was this elimination of records within the same km2 really done only for the Azores? In that case, why (and why use “independent” also in Ln 177 for Europe)? Also, this elimination does not avoid (nor should it) spatial autocorrelation in the presences, but rather in the survey effort.

Ln 177 and 269: Remove “the” before “Europe”.

Ln 204-206: Which were the normal and the non-normal variables? Was Pearson’s coefficient only used when both of the variables in a pair were normally distributed? In any case, this seems like quite an unequal treatment of different variables, as the two coefficients have visibly different power. Do you have a justification or a reference for this procedure?

Ln 223-225: The literature has evolved quite a bit since this reference of 2006, and a few more papers on “best practices” are available nowadays – including Araújo et al. 2019, which is in the reference list although I could not find it cited in the text; and others such as Sofaer et al. 2019 (https://academic.oup.com/bioscience/article/69/7/544/5505326). Also, two important references on Maxent modelling in particular are Elith et al. 2010 (https://onlinelibrary.wiley.com/doi/full/10.1111/j.1472-4642.2010.00725.x) and Merow et al. 2013 (https://onlinelibrary.wiley.com/doi/10.1111/j.1600-0587.2013.07872.x). This should be used to choose appropriate parameters sensibly, rather than just testing limited arbitrary choices.

Ln 226-227: How are 25 and 30% “random test percentages”? Do you mean percentages of 25 and 30% of random test records? I also find this testing of two such similar percentages quite limited, and I find it surprising that a larger proportion of records left out for testing produced apparently better models. This may have to do with both percentages being so similar, and/or with the limited model evaluation procedure (see comment further down).

Ln 228-229: The number of background points is among the main factors affecting model quality, so how did you select this particular number of points, and (especially) why was this parameter not subjected to the sensitivity analysis?

Ln 230: What are the “total background points?”

Ln 234-235: The AUC is not the best metric (especially if used alone) to choose between models, as it has relevant known problems (e.g. Lobo et al. 2008, https://onlinelibrary.wiley.com/doi/abs/10.1111/j.1466-8238.2007.00358.x). Also, the AUC assesses the capacity of models to distinguish between presence and absence records, whereas the data and modelling algorithm in this paper imply presence and background (not absence, nor pseudo-absence) records. Even the reference that the authors cite for “best practices” (Ln 224-25) says that “multiple evaluation measures are necessary to determine accuracy of models produced with presence-only data”.

Ln 235-236: The AUC actually varies between 0 (not 0.5) and 1; 0 means perfect discrimination but backwards classification, and it can happen when models are applied to external sets. Also, 1 does not mean “perfect accuracy”, but perfect discrimination (with correct classification). Accuracy would imply the correctness of the exact continuous values predicted by the model, whereas the AUC only assesses to which extent a model gives higher values (regardless of how much higher) to presence than to absence localities.

Ln 236-239: This sentence is misleading, as some of the parameters mentioned (e.g. number of background points, type of features) were not subjected to sensitivity analysis, so you cannot include them in the “best parameter configuration”. Also, was this configuration selected for all four models tested (M. nivalis and M. putorius in Azores and Europe)?

Ln 271-272: This is only a “validation” if the same field data were not used in model building.

Ln 285-290: Here it is not clear what these model “weights” refer to, and why a weight of 0.86 is better than a weight of 0.99.

Ln 291: “probability of […] occurrence” is not provided by a presence-background model such as Maxent. Do you mean “suitability for […] occurrence”?

Ln 338-341: I don’t see how this is an “ecological” approach. Also, what is relevant is if the proportion of records in areas with suitability >0.5 is higher that expected by chance, given the amount of occurrence records and the amount of pixels available with suitability >0.5. These numbers alone do not prove anything without some assessment of significance, e.g. with a test of equal or given proportions.

Ln 367: Add “more” before “widespread”.

Ln 372-374: This is interesting, but it conflicts with the finding that the introduced range is actually more restricted than would be predicted by the native range, even without these competitors in the introduced range. The whole Discussion should separate and interpret more clearly the results of the different types of models.

Figures 2 and 3: I’d suggest switching to a colourblind-friendly colour scale, as red and green are indistinguishable for a significant fraction of the potential readers.

6. PLOS authors have the option to publish the peer review history of their article (what does this mean?). If published, this will include your full peer review and any attached files.

Reviewer #1: No

Reviewer #2: No

---

## [Author Response · Author response to Decision Letter 0]

4 Mar 2020

University of Azores, 

Angra do Heroísmo, Portugal

04 March 2020

Dear Editor of PLOS ONE

We are pleased to submit the revised version of the manuscript entitled “Modelling the distribution of Mustela nivalis and M. putorius based on native and introduced ranges” for consideration for publication in PLOS ONE. We greatly appreciate the depth of the comments provided by yourself and the reviewers, which we believe enriched and improved the manuscript. Below we detail our response to every comment we have received, how we have incorporated it in the manuscript and the changes we implemented in the new version of the manuscript. We believe that the outcome is a much improved version of the manuscript. Also, we would like considered our manuscript for "Biodiversity Conservation Call for papers" collection.

Looking forward to your assessment,

Kind regards

Lucas Lamelas-López, on behalf of all the co-authors.

ASSOCIATE EDITOR COMMENTS TO AUTHOR:

Thank you for submitting your manuscript to PLOS ONE. After careful consideration, we feel that it has merit but does not fully meet PLOS ONE’s publication criteria as it currently stands. Therefore, we invite you to submit a revised version of the manuscript that addresses the points raised during the review process.

While reviewers agree that this is an important study, there are major concerns about study design and data analysis

Authors’ Response: We have taken both your and the reviewers comments very seriously and incorporated them in the updated version of the manuscript.

JOURNAL REQUIREMENTS:

Authors’ Response: We have revised the style requirements and the file names.

2. Our internal editors have looked over your manuscript and determined that it is within the scope of our Biodiversity Conservation Call for Papers. This collection of papers is headed by a team of Guest Editors for PLOS ONE (https://collections.plos.org/s/biodiversity). The Collection will encompass a diverse range of research articles on biodiversity conservation, including management of invasive species. Additional information can be found on our announcement page: https://collections.plos.org/s/biodiversity

If you would like your manuscript to be considered for this collection, please let us know in your cover letter and we will ensure that your paper is treated as if you were responding to this call. If you would prefer to remove your manuscript from collection consideration, please specify this in the cover letter

 Authors’ Response: Thanks for your suggestion. We would like considered our manuscript for Biodiversity Conservation collection.

3. In your Methods section, please provide additional location information of the study area, including geographic coordinates for the data set if available.

Authors’ Response: We have created a new supplementary material (S2 File), including the geographic coordinates of the occurrence records, according your suggestion and the reviewers comments.

REVIEW COMMENTS TO THE AUTHOR:

REVIEWER #1

GENERAL COMMENTS:

This paper models habitat suitability in the Azores archipelago for two introduced mustelid species, Mustela nivalis and Mustela putorius. For each species, the authors fit two models: (i) one model fitted to a dataset of detections collected over the whole Europe (native range), and (ii) one model fitted to a dataset of detections collected only in the Azores islands (introduced range). For each species, the authors then compare the maps predicted by the two models (introduced and native), and show that the maps obtained with the two datasets are different. The map predicted using data collected over the whole Europe does not correspond to the map predicted using data collected in the Azores. The authors conclude that this difference indicates that species distribution models should be carefully interpreted when used to predict the range of a species in an area based on data collected in another area.

As a whole, I found this paper well written. Their aim is interesting and I fully endorse their conclusion, i.e. that "factors that influence species distribution in the introduced range could be novel or differ from those in their native ranges (...). The difference might even be starker when the introduced ranges include islands while the native ranges are continental areas. Invasive species SDMs can be useful for the management of biological invasions but a careful interpretation is necessary and must be based on ecological knowledge".

Authors’ Response: We appreciate the reviewers’ interest on the aim of our research. We are grateful for your constructive comments and suggestions that helped us to improve the manuscript quality. Below we answer to each of them individually.

However I have concerns related to the statistical methods used in their paper. In addition, I think that this paper should contain more details on the approach used to fit the models, possibly given in supplementary data. It is presently difficult to judge of the validity of the analyses based on the information provided in the paper.

Authors’ Response: Thank you for your concerns regarding the statistical analyses. We agree that there was restricted description of the approach used to fit the models and to amend this we have now included new supplementary materials (Files S1, S2, S5, S6 and S7) that detail the approach step-by-step. We believe that with this information you will be able to assess the validity of the analyses and results presented in the paper. We also included eight new references suggested by the reviewers to support our Methods and Discussion. 

 My main concerns are:

****

My first main concern is related the difference of scales of their two models (native and introduced). The concept of scale is an essential concept in ecology, and we should not expect to obtain the same results at different scales. For example, on a large scale, a savannah ungulate species might select areas close to water during the dry season, but on a small scale rarely use it rarely of the abundance of nearby predators. Thus large scale SDM might predict a high HSI for areas close to water, whereas small-scale SDM might predict low HSI for such areas. 

Authors’ Response: Thank you for your concern and example on how the effect of scale could affect our results. However, we believe this effect is not affecting the model outputs because we used the same spatial resolution and the same data sources for both study areas; therefore, if one factor at a smaller spatial scale is missed it will be missed on both analyses. The only factor that we anticipate to be affected by the different study area sizes is prevalence (i.e. the fraction of the study area that the species can occupy), but this is compensated for by having a similar sampling density for both areas.

Similarly, I do not expect that differences between the distribution of a species over a whole continent with a very large diversity of climates, landscapes, human densities, etc. and the expected distribution in a set of small islands with a comparatively homogeneous climate and landscape to be automatically caused by the fact that the species is native in one place and introduced in another. This difference of scale may for example explain why "native" models predict wider range than "introduced" models. In my opinion, it would be more sensible to compare the Azores to areas with similar size and environmental characteristics (or at least as similar as possible) where the species is natively present.

Authors’ Response: Thank you for your interesting suggestion. We took it into consideration and we concluded that it would be difficult to justify selecting sub areas within continental Europe that would match the size of the Azores to conduct the analysis as suggested. This is because there could be an even bigger effect due to this selection than the potential effects of modeling a smaller area using information from a larger area. Further, we believe that this suggestion is not in line with other approaches to model invasive species ranges. Several authors have also modeled invasive species range from information retrieved from the native range (e.g., Broennimann et al., 2007; Fitzpatrick et al., 2007; Rödder et al., 2009; Bidinger et al., 2012); This is because invasive species are yet to be in equilibrium with their environment in the invaded ranges thus becoming even more important to include a broader set of parameter ranges from the native range to predict areas with potential for invasion. 

Bidinger K, Lötters S, Rödder D, Veith M. Species distribution models for the alien invasive Asian Harlequin ladybird (Harmonia axyridis). J Appl Entomol. 2012; 136(12): 109–123.

Broennimann O, Treier UA, Müller‐Schärer H, Thuiller W, Peterson AT, Guisan A. Evidence of climatic niche shift during biological invasion. Ecol Lett. 2007; 10(8): 701−709.

Fitzpatrick MC, Weltzin JF, Sanders NJ, Dunn RR. The biogeography of prediction error: why does the introduced range of the fire ant over‐predict its native range?. Global Ecol Biogeogr. 2007; 16(1): 24−33.

Rödder D, Schmidtlein S, Veith M, Lötters S. Alien invasive slider turtle in unpredicted habitat: a matter of niche shift or of predictors studied? PLOS ONE. 2009; 4(11).

****

Another important concern is the use of a very heterogeneous detection dataset to fit the models. This dataset was built from various datasources including mainly GBIF and citizen science programs. 

Authors’ Response: Indeed, but this is also the case of GBIF and other datasets commonly used in species distribution models.

As rightly noted by the authors, such data are characterized by numerous collection bias. Thus, species detection are generally more numerous in areas where there are more observers. For example, considering the Terceira island in Fig. 2 and 3, it is clear that most detections occur in places where there are most people (Angra do Heroismo on the Eastern side, Praia da Vitoria on the southern side). The same is true for data collected in mainland Europe.

Thus, any model not accounting for these bias will be biased and predict high suitability indices in areas where there are both many observers and many animals. The only possible approach to SDM with such data implies a model of the spatial distribution of the survey effort.

Authors’ Response: We have noted that SDM datasets are often affected by several bias due to differences in numbers of observers, experience, location, detectability, date, season, etc. Most of these biases resulting from opportunistic sampling may influence SDM performance and predictability. This is not an unknown or unresolved problem in SDM, and to account for these sampling biases we followed the suggestions by Phillips et al. (2009) to chose background data exhibiting the same spatial bias as the presence records. We have included two new Supplementary Material files with all occurrence records used in the study (S2 File), including information about the source (Open Source or Field Data); S1 File contains the methodology used to create the biased background points (see for more details).

In the case of the introduced-range records, most of them were collected from a set of standardized sampling campaigns across all island habitats, therefore without a bias towards urban areas. The authors were involved in these sampling campaigns (e.g., Mathias et al., 1998; Collares-Pereira et al., 2000; Lamelas-López & Salgado, in press). 

Collares-Pereira M, Mathias ML, Santos-Reis M, Ramalhinho MG, Duarte-Rodrigues P. Rodents and Leptospira transmission risk in Terceira island (Azores). Eur J Epidemiol. 2000; 16(12): 1151−1157.

Lamelas-López L &Salgado I (In press). Applying camera-trapping to detect and monitor introduced mammal species on oceanic islands. Oryx.

Mathias MDL, Ramalhinho MG, Santos-Reis M, Petrucci-Fonseca F, Libois R, Fons R, et al. Mammals from the Azores islands (Portugal): an updated overview. Mammalia. 1998; 62(3): 397−408.

Phillips SJ, Dudík M, Elith J, Graham CH, Lehmann A, Leathwick J, Ferrier S. Sample selection bias and presence‐only distribution models: implications for background and pseudo‐absence data. Ecol Appl. 2009; 19(1): 181−197.

The authors acknowledge this point and indicate "presence records from these datasets may be affected by sampling bias, because they are often spatially biased toward easily accessed areas. This spatial bias can lead environmental bias given that the background data are usually drawn at random from the entire region [48]. Consequently, and according to Phillips et al. [48], we chose background data exhibiting the same bias as the presence records." However, the authors do not detail how they chose such biased background data.

It seems that the authors have used the *target group* method of Phillips et al., as they cite this paper. This approach rely on the definition of a broad set of many species, chosen to represent the specimen collection or observation activities of collectors of the target species (so-called "target groups"). Is this indeed the method used here ? If this is the case, what are the species chosen for this target group ? 

Authors response: We have included a new supplementary material containing the methodology used to create the biased background points (See S1 File). The chosen species were our two target invasive mammals, Mustela putorius and Mustela nivalis.

Are the data selected from the same source?

Authors response: Yes. The data was selected by mimicking the sampling bias in the presence data. We included a new ESM which contains the explanation on how this dataset was created and a map with the biased background points used to model the native area (see S1 File - Figure S1). 

Moreover, the Azorean data were "cleaned" by selecting one record per squared km "to avoid a potential bias of spatially auto-correlated presences". I do not see how the method of Phillips et al. could be used with such a treatment: either (i) the same "cleaning" treatment is carried out on the background data (keeping one background point per squared km) and in this case, we loose the bias correction in the model (since the greater probability to find a detection at a given place is no longer reflected by the amount of background points at this place), or (ii) the same "cleaning" treatment is not carried out on the background data, and these data "overcorrect" the bias where the species is detected (since the presence data have a "cleaning" treatment that the background data do not have, and the species can only be detected once at a given place).

Authors response: We performed two corrections (i) sample cleaning, and (ii) spatial biases. Sampling cleaning refers to selecting a grid cell only once to avoid pseudo-replication due to high occurrence within that grid cell or the same animal being recorded more than once within the cell. This is common practice to avoid duplicate cells that inflate model performance while not adding explanatory power. This practice is common in SDM and any other statistical analyses that require independent (or near-independent) data as input. The second correction was the spatial biases because of open data, as addressed in the previous point. The Phillips et al. (2009) method was applied after the trimming to 1km cells being represented only once so it was not affected by this data quality control method. 

Phillips SJ, Dudík M, Elith J, Graham CH, Lehmann A, Leathwick J, Ferrier S. Sample selection bias and presence‐only distribution models: implications for background and pseudo‐absence data. Ecol Appl. 2009; 19(1): 181−197.

I would like to see a map of the distribution of the points of other species used as background data, to have an idea of the spatial distribution of the survey effort. 

Authors Response: We provide this in a new electronic supplementary material (S1 File).

This is an important aspect for a SDM, as the fitting of SDM to such a biased dataset will focus only on parts of geographic space that contain presence samples. As noted by Phillips et al. "predictions into unsampled areas, especially those with conditions outside the range observed in sampled areas, should be treated with strong caution." This is an important warning given the aim of the paper: given the small sample size of data available in the Azores island, I fear that there are areas that are not sampled at all, and this map would allow to identify which predictions should not be trusted at all.

Authors Response: We agree with your point and have acted carefully in our modelling approach to avoid predictions into no-sampled areas. As mentioned above all the island habitats were sampled and we used quality control techniques to avoid pseudo-replication of data and spatial biases. We included this information in the new version of the manuscript. In addition we created uncertainty in prediction maps (S7 File) based on the introduced range by overlapping n-1 models (n= number of presence records). This allows us to determine the consensus, i.e., how many times a given cell was predicted to be suitable (HSI>0.5) for both study species. These two approaches together should give the context of the results and the reliability of the predictions. This is also quite unique in many SDM studies that often do not produce such consensus map that depicts uncertainty.

****

Other methodological concerns are related to the small sample size of the data set used for the "introduced models":

* Note that because the "introduced" model is fitted on a very small dataset (less than 30 detections), the number of variables in the model will necessarily be much smaller than the number of variables in the "native" model based on a much larger dataset (the sample size has a very strong effect on the number of variables that the AIC will select). This difference of sample sizes makes difficult the comparison of the structure of the two separate models (in particular, it is difficult to conclude anything concerning the fact that a particular variable is present in the "native" model and not in the "introduced" model).

Authors’ Response: Indeed the difference in sample sizes could affect how AIC makes model selection. Nonetheless these comparisons are done for each study area individually, i.e., comparing models with similar samples sizes with each other. Thus, we believe that this does not affect which variables would be selected and how similar are they between the island and the continental study area. We would argue they are two independent processes of variable selection that we then compare, having in mind the sampling size effects when making claims of the similarity of the variables for each study area. 

* Visual comparisons of maps in figure 2 and 3 are actually simply visual comparisons of point predictions by models. The prediction uncertainty is not accounted for in theses comparisons. Given the small sample size for "introduced" models, and given that some areas in the Azores islands seem to be scarcely sampled, this will probably lead to weak conclusions at these places.

Authors’ Response: Although we have a small dataset for the introduced area, some studies revealed that MaxEnt performs well in these cases (e.g., around 25 records as in our case; Hernandez et al., 2006; Bean et al., 2012; Proosdij et al., 2016). Further the data we use comes from sampling sessions that covered all the habitats in the island, reflecting the total existing data for the Azores, in fact currently there are no more available records in the Azores that comply with our selection criteria. However, the model predictions to remote or inaccessible areas were interpreted with caution (see Discussion section for more details).

Bean WT, Stafford R, Brashares JS. The effects of small sample size and sample bias on threshold selection and accuracy assessment of species distribution models. Ecography. 2012; 35(3): 250−258.

Hernandez PA, Graham CH, Master LL, Albert DL. The effect of sample size and species characteristics on performance of different species distribution modeling methods. Ecography. 2006; 29(5): 773−785.

Proosdij AS, Sosef MS, Wieringa JJ, Raes N. Minimum required number of specimen records to develop accurate species distribution models. Ecography. 2016; 39(6): 542−552.

*****

SPECIFIC COMMENTS:

* Line 241--247. The description of the models is not clear. Would it be possible to present them in a table ? Presently, it is very difficult to understand which models have been compared. For example, the second model presented here is: "topographic + climatic + landscape variables (n=9) + topographic + climatic + human variables (n=8)". Here, "Topographic + climatic" is present twice ? I think that a more detailed explanation is required.

Authors’ Response: We agree with your comment. We have included a new supplementary material (S6 File) to clarify this. In the example that you indicate, we made a mistake: we should be included a comma between (n=9) and + topographic, because are two different sets of variables. We have corrected this, thank you.

* Table 2: for the model "M. putorius, introduced": what is the difference between the two models since both include the same variables?

Authors’ Response: We have corrected this Table, including the referred comment.

* line 282 : "For M. nivalis, one model was selected for both the native and the introduced range". Do the authors mean "one model was selected for each dataset" ? The selected model is indeed not the same for the two species.

Authors’ Response: We have simplified this text according to comments from both reviewers (Results - Model selection). 

* line 281--290. It is not necessary to include the Akaike weights both in the table and in the text.

Authors’ Response: Thank you for the suggestion, we removed the Akaike weight from the text.

* Table 4: there are several problems with this table, for the "introduced" models: the sum of percentage for M. putorius is >100%, and for Mustela nivalis, > 50% of the records are characterized with HSI<0.5, contrarily to what is written in the text.

Authors’ Response: You are right. We have corrected Table 4 values and included in the manuscript the particular case of M. nivalis introduced-range HSI values.

REVIEWER #2

GENERAL COMMENTS:

This a well-written and potentially interesting study modelling two species’ distributions based on native and (insular) introduced ranges. However, some parts of the methods need further explanation or tuning, and the Discussion also needs improvements. It is also not clear where the species occurrence data collected by the authors are made publicly available.

Authors’ Response: Thank you for your suggestions and appreciation for our findings. We appreciate your highly constructive comments that helped us to improve the quality of the manuscript, particularly the Methods and Discussion section. Additionally, we have included new five supplementary files and eight new references to support these sections. 

Figure 1 shows a substantial problem of survey bias, with some countries presenting a very high and other countries a very low density of occurrence records, without this reflecting the actual occurrence patterns of the analysed species. France and Switzerland are very clear examples of countries providing wildly insufficient data in this case, but even other countries like Spain and Portugal are clearly under-represented in the analysed dataset, compared to other data sources (at broader spatial resolutions) such as the national mammal atlases, which show that these species occur in many more areas than were used in the native range models. Moreover, this bias is not related to accessible areas, and it is unclear how the Maxent bias analysis deals with it.

Authors’ Response: Thanks for the comment. We agree with you that occurrence records from open source data are often affected by sampling bias. Phillips et al. (2009) proposed to select background data exhibiting the same bias as the presence data. For example, if the presence data are taken of a determinate portion of the study area, then the background data should be taken from the same areas. We have included a new supplementary material describing how we did this background-data bias process (S1 File). Additionally, we re-wrote the sentence about records bias. 

Phillips SJ, Dudík M, Elith J, Graham CH, Lehmann A, Leathwick J, Ferrier S. Sample selection bias and presence‐only distribution models: implications for background and pseudo‐absence data. Ecol Appl. 2009; 19(1): 181−197.

Other parts of the methods also need better explanation and justification, such as the limited variations in model parameters that were chosen for sensitivity analysis, as well as the limited method for model evaluation and selection. Recent specific literature on good modelling practices should be used for better tuning of model parameters.

Authors’ Response: We have improved the Methods - Parameter configuration section, and included some relevant references (see the answers to the specific comments below). 

Figures 2 and 3 show very different (sometimes nearly opposite) predictions from native-range and introduced-range models. This needs to be explored and explained more clearly – and separately – in the Discussion.

Authors’ Response: We agree with your comment and, accordingly, we have revised and, whenever needed, modified the discussion to better emphasize the results from the different analytical approaches and by explaining more clearly our interpretation of the obtained results, especially the surprising ones. 

SPECIFIC COMMENTS:

Ln 20: I would add “archipelago” after “Azores”, as this is a relevant biogeographic feature which may not be immediately obvious to all readers.

Authors’ Response: We added “archipelago” after “Azores".

Ln 24-26: The current phrasing is confusing as to whether the “differences” are between species or between models. I suggest rephrasing to something like “We found differences in the predicted distributions of models based on introduced and native occurrences for both M. nivalis and M. putorius in the Azores”.

Authors’ Response: We changed the sentence according to your suggestion.

Ln 32: It its not clear what “this” refers to.

Authors’ Response: We removed this sentence from the abstract in the new version of the manuscript.

Ln 62: The Azores are more than one island, so “island’s ” should be “islands’ ”

Authors’ Response: We changed "island’s" to "islands’"

Ln 55-59: This paragraph refers common mammals in the Azores, but most of the provided references are focused on birds. I miss a reference to the atlas of Portuguese mammals, which seems to be mentioned in acknowledgments but is not among the cited literature.

Authors’ Response: We included the suggested reference in the manuscript.

Ln 69: “distributions” should be “distribution”.

Authors’ Response: We changed the text accordingly.

Ln 78: Overly long sentence. I would add a comma after “conservatism”.

Authors’ Response: We changed the text accordingly.

Ln111: Replace the final comma with a full stop and start a new sentence.

Authors’ Response: Done.

Ln 144: “deliberately” should be “deliberate”.

Authors’ Response: We have changed the text accordingly.

Ln 145: “or accidentally” is redundant in this sentence.

Authors’ Response: We removed "or accidentally" from the text.

Ln 163: Were these direct observations and sampling campaigns conducted across the modelled range, or centered on particular regions/countries? It would also be interesting to mention how many points were added by the authors to the data available in the open published datasets, and if these data were added to these (or to which) public platforms.

Authors’ Response: We specified the country where we collected the data, according your comment. Also, we included a new supplementary material (S2 File) with all presence records used in the study. We detailed if the record was obtained from open source database or was collected by the authors.

Ln 164: How was this record accuracy assessed? Namely, which columns of the public databases were used for this filtering, and with which values?

Authors’ Response: Most of the records were obtained from the GBIF database. We downloaded the records as a .csv file from the GBIF platform. Then we filtered the accuracy of the record from "coordinateUncertaintyInMeters" column. The records with values equal or higher than 1000 meters were removed. Also, we removed the records without information in this column. We clarify this in the Methods section - Species presence data.

Ln 166: In this case, the bias was clearly not (only) towards accessible areas, but also reflected the habits of different countries in uploading occurrence records to the analysed databases. France and Switzerland, for example, have a lot of accessible areas but a tiny portion of records.

Authors’ Response: We have included in the new version this irregular facilitation/uploading of the occurrence records (Methods section - Species presence data). 

Ln 168-169: More details are necessary on how this biased background sample was generated exactly, and a map of these background points should be included either in the article or the supplementary files.

Authors’ Response: We created a new Supplementary Material (S1 File), explaining how were selected the biased background points, according your comment.

Ln 169: I would start a new paragraph at “The introduced...”

Authors’ Response: We have changed the text accordingly.

Ln 173-174: Was this elimination of records within the same km2 really done only for the Azores? In that case, why (and why use “independent” also in Ln 177 for Europe)? Also, this elimination does not avoid (nor should it) spatial autocorrelation in the presences, but rather in the survey effort.

Authors’ Response: You are right. We have selected only one record per 1km2 in both ranges. We have re-written the Methods - Species presence data Section to include this.

Ln 177 and 269: Remove “the” before “Europe”.

Authors’ Response: Done.

Ln 204-206: Which were the normal and the non-normal variables? Was Pearson’s coefficient only used when both of the variables in a pair were normally distributed? In any case, this seems like quite an unequal treatment of different variables, as the two coefficients have visibly different power. Do you have a justification or a reference for this procedure?

Authors’ Response: Thanks for your comment. We newly performed a normality test and detected a mistake in the correlation analysis because all variables are non-normally distributed; so we repeated the correlation analysis and consequently used the Spearman rank coefficient (see S4 File for the new results). 

Ln 223-225: The literature has evolved quite a bit since this reference of 2006, and a few more papers on “best practices” are available nowadays – including Araújo et al. 2019, which is in the reference list although I could not find it cited in the text; and others such as Sofaer et al. 2019. Also, two important references on Maxent modelling in particular are Elith et al. 2010 and Merow et al. 2013. This should be used to choose appropriate parameters sensibly, rather than just testing limited arbitrary choices.

Authors’ Response: Thanks for this constructive comment. We have included the suggested references in the text.

Ln 226-227: How are 25 and 30% “random test percentages”? Do you mean percentages of 25 and 30% of random test records? I also find this testing of two such similar percentages quite limited, and I find it surprising that a larger proportion of records left out for testing produced apparently better models. This may have to do with both percentages being so similar, and/or with the limited model evaluation procedure (see comment further down).

Authors’ Response: We agree with you in that to test two percentages very similar is quite limited. For this reason we removed this aspect of the testing process.

Ln 228-229: The number of background points is among the main factors affecting model quality, so how did you select this particular number of points, and (especially) why was this parameter not subjected to the sensitivity analysis?

Authors’ Response: We have selected around 30% of the total background points for each area (1,800 for Azores and 10,000 for Europe). Some authors recommend the use a large number of background points (e.g. 10,000); however, for our case we chose only 30% of these values and chose to run models 10 times, according Barbet-Massin et al. 2012. We used also this lower number of background points as a trade-off between the number of different combinations of models and the time for model runs.

Barbet‐Massin M, Jiguet F, Albert CH, Thuiller W. Selecting pseudo‐absences for species distribution models: how, where and how many?. Methods Ecol Evol. 2012; 3(2): 327−338.

Ln 230: What are the “total background points?”

Authors’ Response: We have removed this sentence in the new version of the manuscript.

Ln 234-235: The AUC is not the best metric (especially if used alone) to choose between models, as it has relevant known problems (e.g. Lobo et al. 2008). Also, the AUC assesses the capacity of models to distinguish between presence and absence records, whereas the data and modelling algorithm in this paper imply presence and background (not absence, nor pseudo-absence) records. Even the reference that the authors cite for “best practices” (Ln 224-25) says that “multiple evaluation measures are necessary to determine accuracy of models produced with presence-only data”.

Authors’ Response: We used AUC only to test the different parameter configuration performance (see Methods - Parameter configuration). Then, when parameter configuration was selected, we performed model selection using AICc (see Methods - Model selection).

Ln 235-236: The AUC actually varies between 0 (not 0.5) and 1; 0 means perfect discrimination but backwards classification, and it can happen when models are applied to external sets. Also, 1 does not mean “perfect accuracy”, but perfect discrimination (with correct classification). Accuracy would imply the correctness of the exact continuous values predicted by the model, whereas the AUC only assesses to which extent a model gives higher values (regardless of how much higher) to presence than to absence localities.

Authors’ Response: You are right. We have corrected the text accordingly.

Ln 236-239: This sentence is misleading, as some of the parameters mentioned (e.g. number of background points, type of features) were not subjected to sensitivity analysis, so you cannot include them in the “best parameter configuration”. Also, was this configuration selected for all four models tested (M. nivalis and M. putorius in Azores and Europe)?

Authors’ Response: We re-wrote the sentence. Yes, this configuration was used for all four models tested.

Ln 271-272: This is only a “validation” if the same field data were not used in model building.

Authors’ Response: We have used a subset of presence data randomly selected to perform the validation. We have clarified this in the main text.

Ln 285-290: Here it is not clear what these model “weights” refer to, and why a weight of 0.86 is better than a weight of 0.99.

Authors’ Response: We have modified the text in the new version of the manuscript (Results - Model selection). 

Ln 291: “probability of […] occurrence” is not provided by a presence-background model such as Maxent. Do you mean “suitability for […] occurrence”?

Authors’ Response: We have replaced "probability" by "suitability".

Ln 338-341: I don’t see how this is an “ecological” approach. Also, what is relevant is if the proportion of records in areas with suitability >0.5 is higher that expected by chance, given the amount of occurrence records and the amount of pixels available with suitability >0.5. These numbers alone do not prove anything without some assessment of significance, e.g. with a test of equal or given proportions.

Authors’ Response: Thanks for your suggestion. We have included a Chi2 test to evaluate if the HSI values were significantly different than expected by chance, as suggested.

Ln 367: Add “more” before “widespread”.

Authors’ Response: Done.

Ln 372-374: This is interesting, but it conflicts with the finding that the introduced range is actually more restricted than would be predicted by the native range, even without these competitors in the introduced range. The whole Discussion should separate and interpret more clearly the results of the different types of models.

Authors’ Response: Done.

Figures 2 and 3: I’d suggest switching to a colourblind-friendly colour scale, as red and green are indistinguishable for a significant fraction of the potential readers.

Authors’ Response: Thanks for your suggestion. We have changed the scale colour of the Figures 2 and 3.

---

## [Decision Letter · Decision Letter 1]

17 Apr 2020

Your manuscript has the potential to provide useful information. However, it is important that you carefully read and respond to the concerns and detailed explanations from Reviewer 1.

Dear Mr Lamelas-López,

Thank you for submitting your manuscript to PLOS ONE. After careful consideration, we feel that it has merit but does not fully meet PLOS ONE’s publication criteria as it currently stands. Therefore, we invite you to submit a revised version of the manuscript that addresses the points raised during the review process.

We would appreciate receiving your revised manuscript by Jun 01 2020 11:59PM. To enhance the reproducibility of your results, we recommend that if applicable you deposit your laboratory protocols in protocols.io, where a protocol can be assigned its own identifier (DOI) such that it can be cited independently in the future. For instructions see: http://journals.plos.org/plosone/s/submission-guidelines#loc-laboratory-protocols

We look forward to receiving your revised manuscript.

Kind regards,

Ulrike Gertrud Munderloh, Ph.D.

Academic Editor

PLOS ONE

Reviewers' comments:

Reviewer's Responses to Questions

**Comments to the Author**

1. If the authors have adequately addressed your comments raised in a previous round of review and you feel that this manuscript is now acceptable for publication, you may indicate that here to bypass the “Comments to the Author” section, enter your conflict of interest statement in the “Confidential to Editor” section, and submit your "Accept" recommendation.

Reviewer #1: (No Response)

Reviewer #2: All comments have been addressed

2. Is the manuscript technically sound, and do the data support the conclusions?

Reviewer #1: No

Reviewer #2: Yes

3. Has the statistical analysis been performed appropriately and rigorously? 

Reviewer #1: No

Reviewer #2: Yes

4. Have the authors made all data underlying the findings in their manuscript fully available?

Reviewer #1: Yes

Reviewer #2: Yes

5. Is the manuscript presented in an intelligible fashion and written in standard English?

Reviewer #1: Yes

Reviewer #2: Yes

6. Review Comments to the Author

Reviewer #1: This paper is a review of a previous paper submitted to PLOS One. Its aim is to model habitat suitability in the Azores archipelago for two introduced mustelid species, Mustela nivalis and Mustela putorius. For each species, the authors fit two models: (i) one model fitted to a dataset of detections collected over the whole Europe (native range), and (ii) one model fitted to a dataset of detections collected only in the Azores islands (introduced range). For each species, the authors then compare the maps predicted by the two models (introduced and native), and show that the maps obtained with the two datasets are different. The map predicted using data collected over the whole Europe does not correspond to the map predicted using data collected in the Azores. The authors conclude that this difference indicates that species distribution models should be carefully interpreted when used to predict the range of a species in an area based on data collected in another area.

In my review of the previous manuscript, I expressed several major concerns related to the methods used to reach the aim, as well as several minor comments. Many of my comments have been satisfactorily taken into account. However, I think that we disagree on my main comment: I think that the comparison between native and introduced models carried out in this paper does not allow to draw conclusions on the ability to predict introduced range from the native range, because of the very different scales of study for the two areas.

It is possible that I was not clear enough, so that I explain more clearly below why I think that the metholodogy in this paper is very problematic. Moreover, the more precise description that the authors now give of the methodology used to correct for sampling bias raises another major methodological problem.

I describe these two problems below.

******

** On the concept of scale.

In my previous review, my main criticism was related to the difference of geographical scales between the model fitted in the native range and the model fitted in the Azores archipelago. I expressed it in my previous review by noting that results may vary strongly between scales, and comparing a model fitted using data collected over continental scale and a model collected on a higly local scale is meaningless, as the two models are necessarily describing very different processes, so that they are necessarily returning different results.

To this criticism, the authors replied: "we believe this effect is not affecting the model outputs because we used the same spatial resolution and the same data sources for both study areas."

Actually, the concept of scale in Ecology is more general than just the resolution of the study (Dungan et al. 2002): the concept of scale involves many aspects (grain, lag, support, etc.). But in this study, the most problematic aspect is the extent of the study area, rather than its resolution. Dungan et al. (2002) illustrates clearly how the results of a study might change strongly when the scale changes. Another reference here -- a seminal one actually -- is Johnson (1980), who defines in the discussion four orders of habitat selection (first order: geographical range, second order: distribution of animals in a region, third order: distribution of animal locations within their home ranges, fourth order: selection of food items at the locations). The processes and preferences at a scale are not necessarily the same than those at another. In the present study, the authors compare first order selection (whole Europe) with second order selection (distribution within the islands). Many other authors have stressed the importance of this scale and variability of the results according to the scale considered (e.g. Pearce and Boyce 2006, Soberon and Peterson 2005).

More concretely, here, any analysis of environmental variables classically used for SDM on the continental scale in Europe will mainly show large environmental patterns. For example, on a continental scale, the effect of elevation on the presence of a species will be driven by the differences between mountainous areas (the Alps) and other areas. That is, by the differences of weather, climate, vegetation, snow cover, etc. between Alpine and non-Alpine climates. Of course, other climatic variables will also account for it to some extent, but no variable will synthetize the difference between non-Alpine and Alpine climate as efficiently as the elevation (by definition !). Therefore, any effect of the elevation in SDM models at this scale will summarize the difference of environments (vegetation, climate, etc.) between Alpine and non-Alpine environments. Therefore, if -- for example -- a species is absent or rarer in the Alps than in the rest of the continent, it will be because the Alpine climate is less suitable for the species than non-Alpine climates (whatever the reason, e.g. too much snow in winter, rocky soils, etc.). This will be synthetized by a negative effect of the elevation in the SDM.

On the other hand, the elevation, in the Azores archipelago, has a very different meaning. In these islands, high elevation areas are not characterized by a "more Alpine climate" than other areas. It has a very different meaning. In these islands, for example, there is probably a very strong correlation between elevation and the distance to the sea. So that elevation here rather represents this distance (e.g. further from the sea = less urban, etc. -- maybe a better habitat for the species?). Thus the meaning of elevation for the species is not the same, just because we do not work at the same scale. Of course, my example here describes a fictious species (I do not know what are the selection patterns for weasel and ferret at this scale), but it illustrates why results are expected to vary between two scales, even within the native range (e.g. the results will not be the same if we compare a model fitted on the whole Europe and an model fitted e.g. on similar sized areas in continental Portugal).

The same is true for e.g. climatic variables. The effect of these variables on a continental scale will likely reflect the differences in densities between the different climates in Europe. If a species is less abundant in e.g. areas with continental climates (with less precipitation), then the effect of clim_bio12 in a SDM on a continental scale will mainly represent this large scale pattern. At the scale of the Azores Islands, this variable will not have the same meaning, as there is no continental climate in the Azores islands. Etc.

If the aim is to predict the introduced range from data collected in the native range, it would make much more sense to compare this archipelago with areas *of similar sizes* located in similar climates, with a similar elevation range, etc.

I suggested this approach in my previous review, but the authors replied:

"We concluded that it would be difficult to justify selecting sub areas within continental Europe that would match the size of the Azores to conduct the analysis as suggested. This is because there could be an even bigger effect due to this selection than the potential effects of modeling a smaller area using information from a larger area."

I do not understand why it would be difficult to justify it. I do not think that there would be a bigger effect in this selection. The authors have a very precise description of their study area according to numerous environmental variables (climate, elevation, etc.). It would be easy to select randomly numerous places of similar sizes in Europe. The standardized environmental variables (i.e. minus the mean and divided by the standard deviation) then each define a dimension in a multidimensional space (one dimension is the average temperature, one is the elevation, on is the precipitation, etc.). Each one of these random areas would define a point in this space. The Azores archipelago also defines a point in this space. Then, we can select a sample of -- say -- 10 random places with the smallest Euclidean distance in this multidimensional space to Azores. The resulting sample will be a random and objective sample of by areas taken in the native range with sizes and environmental conditions similar to the target area. This would be a better approach in my opinion: if the aim is to infer the effect of a factor on a process by comparing two sets of areas -- one with the factor and one without -- it is better to design the study so that only that factor varies and the other variables are identical. Here, to try to find one or several study areas in the native range with sizes and conditions similar to those of the Azores archipelago, with only "introduced/native" differing. On the other hand, as exemplified above, ignoring the effect of scale will lead to erroneous conclusions.

The authors further note:

"Further, we believe that this suggestion is not in line with other approaches to model invasive species ranges. Several authors have also modeled invasive species range from information retrieved from the native range (e.g., Broennimann et al., 2007; Fitzpatrick et al., 2007; Rödder et al., 2009; Bidinger et al., 2012); This is because invasive species are yet to be in equilibrium with their environment in the invaded ranges thus becoming even more important to include a broader set of parameter ranges from the native range to predict areas with potential for invasion."

However, none of these works compare areas with so large size differences. Broenniman et al. (2007) predict the distribution of spotted knapweed in North America with a model fitted with data collected in its native range in western Europe, two areas covering similar sizes. Fitzpatrick et al. (2007) compare the distribution of red ants in tropical south America and tropical north America, again at similar scales and in similar climates. Rödder et al. (2009) study the slider turtle in their native range (North America), and the invasive range is also on a continental scale. Finally, Bidinger et al. (2012) studies the distribution of Harlequin ladybird in their native range in eastern Asia, and in their introduced range (area of similar size in Europe).

I fully endorse the aim of the present study. From a conservation as well as ecological point of view, it is essential to find a way to predict the introduced range from the native range of invasive species. I agree on the importance of understanding how the introduced range might differ from the native range to identify how a species adapt to a new environment. I do not disagree on the aim, but on the method. The problem of scale is not a minor one in ecology. Comparing two areas of very different sizes amounts to compare a process at two very different scales. Therefore, different results are expected, even if there is no difference between native and introduced ranges.

******

** On the correction of sampling effort.

The authors rightly explained that the biased data collection characterizing the dataset would lead to to biased inference if it was not taken into account. They use the "target group" method of Philips et al. (2009) to collect a biased sample of background points exhibiting the same bias as the presence records. The authors did not describe clearly their approach in the previous version of the manuscript. It is now more clearly described.

The "target group" approach of Philips et al. aims at distinguishing whether the absence of detection of the focus species at one place is caused by the absence of the species itself or by the absence of data collection. The idea is to define a "target group" containing many species for which we think that the collection or observation activities of collectors is similar to those of the focus species. For example, to model the SDM of a particular bird species in a citizen science program, it would make sense to use the whole set of bird species studied in the citizen science program as the target group used to select background points. The hope is that, at any point where data collection for the focus species has occurred, at least one species of the target group was present and reported by the same data collection (even if the focus species itself is not). So that the presence of a species of the target group and the absence of the focus species ensures that the absence of reported detection for this focus species is likely due to the actual absence of the species.

This method allows to correct -- to some extent, it is impossible to completely account for all the bias in such contexts -- the collection bias because the target group is usually made of a large number of species, and at least one is supposed to be present where data collection has occurred. In the paper of Philips, for example, the target group contains from 7 to 52 species.

In the present paper, the target group was defined by just the ferret and the weasel. This is a small target group! It means that any area unsuitable for both species, but where data collection has actually occurred will not be present in the data. The background sample will be defined by the set of habitat conditions allowing the presence of the weasel and/or the ferret (well, a biased sample of it, but this is the aim of the target group method to obtain such a biased sample of background points). So that modelling the presence of the weasel (resp. ferret) in a set of locations defined so that either the ferret or the weasel are present, is not a model of the species distribution. It is a model of the niche difference between the two species: the model predicts the probability of presence of one species given that at least one of the two species is present.

So that the "native" model describes the niche differences between ferret and weasel modelled at the scale of the continent species, whereas the "introduced" model describes theses differences at the scale of the archipelago. In other words, the models do not focus on the potential distribution of the species as indicated in the paper, but on the differences between the two species. Which may be of interest, but is not the actual aim of the study.

******

** References

Dungan, J.; Perry, J.; Dale, M.; Legendre, P.; Citron-Pousty, S.; Fortin, M.; Jakomulska, A.; Miriti, M. & Rosenberg, M. 2002. A balanced view of scale in spatial statistical analysis. Ecography 25, 626-640

Johnson, D. 1980. The comparison of usage and availability measurements for evaluating resource preference. Ecology 61, 65-71

Levin, S. 1992. The problem of pattern and scale in Ecology. Ecology 73, 1943-1967

Soberon, J. & Peterson, A. 2005. Interpretation of models of fundamental ecological niches and species' distributional areas. Biodiversity Informatics 2, 1-10

Pearce, J. & Boyce, M. 2006. Modelling distribution and abundance with presence-only data. Journal of Applied Ecology 43, 405-412

Elith, J. & Leathwick, J. R. 2009. Species distribution models: ecological explanation and prediction across space and time. Annual Review of Ecology, Evolution, and Systematics 40, 677

Reviewer #2: The authors have addressed the concerns expressed in the previous review. The explanation and map provided in Appendix S1 make the procedure much clearer and more transparent, and the dataset provided in appendix S2 adds significant value and usefulness to the manuscript. All other supplementary materials also help make the methodology clearer. Although not everything was done the way I would have done it myself, I am generally satisfied with the current version of the manuscript and I believe the authors have appropriately explained and defended their work.

7. PLOS authors have the option to publish the peer review history of their article (what does this mean?). If published, this will include your full peer review and any attached files.

Reviewer #1: No

Reviewer #2: No

---

## [Author Response · Author response to Decision Letter 1]

6 Jul 2020

Response to Reviewers

Reviewer #1: 

This paper is a review of a previous paper submitted to PLOS One. Its aim is to model habitat suitability in the Azores archipelago for two introduced mustelid species, Mustela nivalis and Mustela putorius. For each species, the authors fit two models: (i) one model fitted to a dataset of detections collected over the whole Europe (native range), and (ii) one model fitted to a dataset of detections collected only in the Azores islands (introduced range). For each species, the authors then compare the maps predicted by the two models (introduced and native), and show that the maps obtained with the two datasets are different. The map predicted using data collected over the whole Europe does not correspond to the map predicted using data collected in the Azores. The authors conclude that this difference indicates that species distribution models should be carefully interpreted when used to predict the range of a species in an area based on data collected in another area.

In my review of the previous manuscript, I expressed several major concerns related to the methods used to reach the aim, as well as several minor comments. Many of my comments have been satisfactorily taken into account. However, I think that we disagree on my main comment: I think that the comparison between native and introduced models carried out in this paper does not allow to draw conclusions on the ability to predict introduced range from the native range, because of the very different scales of study for the two areas.

It is possible that I was not clear enough, so that I explain more clearly below why I think that the methodology in this paper is very problematic. Moreover, the more precise description that the authors now give of the methodology used to correct for sampling bias raises another major methodological problem.

I describe these two problems below.

Authors: We are grateful for your detailed review. You have raised two major concerns regarding the methodology: (i) scale and (ii) bias correction. In summary we believe these concerns are not fully warranted because of the way we designed our study, the comparison between islands and continents, the predictor variables selected, and the methods chosen. In general, we followed methodologies, software and analysis described as best practices used in SDM (Araujo et al. 2019). 

Indeed, we agree that SDM transferability should be better studied, which is perhaps extremely important when using these tools for, for example, studies that project range expansions or considerable range contractions – changing dramatically spatial extent of predictor beyond the information range, and invasive species. One way to ameliorate the spatial extent effect could be by having similar sampling intensities, or iteratively transfer models across spatial locations or points in time. This would make for very nice studies, but much beyond our simple analysis. Below we provide much more detail to our reasoning for the issue of scale and bias correction.

Araújo MB, Anderson RP, Barbosa AM, Beale CM, Dormann, C. F., Early, R., et al. Standards for distribution models in biodiversity assessments. Sci Adv. 2019; 5(1): eaat4858.

******

** On the concept of scale.

In my previous review, my main criticism was related to the difference of geographical scales between the model fitted in the native range and the model fitted in the Azores archipelago. I expressed it in my previous review by noting that results may vary strongly between scales, and comparing a model fitted using data collected over continental scale and a model collected on a highly local scale is meaningless, as the two models are necessarily describing very different processes, so that they are necessarily returning different results.

Authors: Thank you for your comment, this is indeed a major aspect to consider in any SDM study. Soberon (2007) illustrated that species respond to environmental conditions like habitat type, food resources, etc at local scales, and to environmental conditions like bioclimate and topography at larger scales and that in order to fully understand factors that drive species ranges we would need to derive nested models at these scales. Unfortunately, the literature on SDM has yet to get to this stage but many examples are now emerging that combine both bioclimatic and habitat variables and show that there is an interplay of the importance of the two (Santos et al. 2017). This is further complicated when dealing with invasive species which likely change both associations with habitat and bioclimatic parameters in the invaded range – that is what makes them invasive in the first place. Further, as for our case, there is a complication because we are studying islands. The spatial extent of the invaded area is spatially confined because these are islands, and therefore any conversion of this spatial extent to a continuous continental surface would be arbitrary, and add spurious edge effects for the continental models which would not be present in the islands. To account for all these aspects we carefully designed our study so that:

1. We included the same predictor variables in both the islands and the continental area. These include both bioclimatic variables – to represent processes at large scales, and habitat variables – to represent processes at local scales. In the text: ”We selected (1) two topographic variables (e.g., slope) in order to describe the physical environment; (2) four climatic variables (e.g., annual temperature, precipitation seasonality) to describe the bioclimatic conditions; (3) a set of seven landscape level variables, including habitat type, vegetation structure (herbs, shrubs or trees), land use types, and landscape heterogeneity (e.g., number of patches); and (4) three human disturbance variables, related with human population (e.g., population density) and their activities (% of artificial areas, % of agricultural areas).”

2. We collected the same variables in the islands and the continent. To represent the continent, we did not analyze any continuous surface, but rather chose to select 10,000 isolated cells, randomly selected from whole Europe. 

3. We experimented with the scenario you proposed to restrict the native-area to a small portion of the native range of both species that would match the spatial extent of the Azores. We were confronted with several choices that we were not comfortable making – (i) where to seed the area? (ii) Which shape should it take? (iii) are we inducing arbitrarious edges? (iv) if we created 2,250 km2 areas, how do we then justify the differences between the models? (v) how do we control for very uneven sample size within the Azores, like continental areas? Additionally, the particular environmental conditions of the Macaronesian archipelagos, and particularly of the Azores, are not "comparable" with a continental area with a similar extent. Therefore, we opted to sample from the native-range and used 10,000 isolated cells.

4. We are unaware of where do the species invade from, so this would also limit selecting certain areas within the European range.

5. Of course it is possible that mismatched spatial extents of study areas affect model transferability, however this is an aspect seldom analyzed in SDM studies. Transferability studies are few and far in between, and there are yet no clear guidelines on how to do them properly because indeed we are transferring models developed with a given range of a variable to areas where the range of values the variable can take is different – this is the case for invasive species. Further studies should do an experiment on this topics.

Soberón J. Grinnellian and Eltonian niches and geographic distributions of species. Ecol Lett. 2007; 10(12): 1115−1123.

Santos MJ, Smith AB, Thorne JH. et al. The relative influence of change in habitat and climate on elevation range limits in small mammals in Yosemite National Park, California, U.S.A. Clim Chang Responses. 2017; 4: 7.

To this criticism, the authors replied: "we believe this effect is not affecting the model outputs because we used the same spatial resolution and the same data sources for both study areas."

Actually, the concept of scale in Ecology is more general than just the resolution of the study (Dungan et al. 2002): the concept of scale involves many aspects (grain, lag, support, etc.). 

Authors: We agree and see above our comment to this. Indeed what we are talking about is the extent of the study area.

But in this study, the most problematic aspect is the extent of the study area, rather than its resolution. Dungan et al. (2002) illustrates clearly how the results of a study might change strongly when the scale changes. Another reference here -- a seminal one actually -- is Johnson (1980), who defines in the discussion four orders of habitat selection (first order: geographical range, second order: distribution of animals in a region, third order: distribution of animal locations within their home ranges, fourth order: selection of food items at the locations). The processes and preferences at a scale are not necessarily the same than those at another. In the present study, the authors compare first order selection (whole Europe) with second order selection (distribution within the islands). Many other authors have stressed the importance of this scale and variability of the results according to the scale considered (e.g. Pearce and Boyce 2006, Soberon and Peterson 2005).

Authors: Thank you for your comment, this is indeed a major aspect to consider in any SDM study. As mentioned above, we took the approach from Soberon (2007), which illustrated that species respond to environmental conditions like habitat type, food resources, etc at local scales, and to environmental conditions like bioclimate and topography at larger scales and that in order to fully understand factors that drive species ranges we would need to derive nested models at these scales. Unfortunately, the literature on SDM has yet to get to this stage but many examples are now emerging that combine both bioclimatic and habitat variables and show that there is an interplay of the importance of the two (Santos et al. 2017). This is further complicated when dealing with invasive species which likely change both associations with habitat and bioclimatic parameters in the invaded range – that is what makes them invasive in the first place. 

More specifically, our selected native area is not strictly a continental area, as we are sampling 10,000 isolated cells within Europe (Fig R1 in "Response to Reviewers" Letter; see ESM1 for more details). Therefore, the extent (i.e. the total area) used during the modeling process is similar between regions (extent of introduced range: 2,250 background points, extent of native range: 3,000 background points). As the native area is formed by isolated cells obtained from a continental area, the scale is not necessarily a problem. To address Johnson (1980) orders of habitat selection, we chose variables that represent processes at each of the orders: (1) First order: geographical range – we use bioclimatic variables; (2) Second order: distribution of animals in a region – we use habitat and topographic variables; (3) Third order: distribution of animal locations within their home ranges – vegetation structure (herbs, shrubs or trees), human presence; (4) Fourth order: selection of food items at the locations – vegetation structure. 

Soberón J. Grinnellian and Eltonian niches and geographic distributions of species. Ecol Lett. 2007; 10(12): 1115−1123.

Santos MJ, Smith AB, Thorne JH. et al. The relative influence of change in habitat and climate on elevation range limits in small mammals in Yosemite National Park, California, U.S.A. Clim Chang Responses. 2017; 4: 7.

Phillips SJ, Dudík M, Elith J, Graham CH, Lehmann A, Leathwick J, Ferrier S. Sample selection bias and presence‐only distribution models: implications for background and pseudo‐absence data. Ecol Appl. 2009; 19(1): 181−197.

Ferrier S, Watson G, Pearce J, Drielsma M. Extended statistical approaches to modelling spatial pattern in biodiversity in northeast New South Wales. I. Species-level modelling. Biodivers Conserv. 2002; 11(12): 2275−2307.

More concretely, here, any analysis of environmental variables classically used for SDM on the continental scale in Europe will mainly show large environmental patterns. For example, on a continental scale, the effect of elevation on the presence of a species will be driven by the differences between mountainous areas (the Alps) and other areas. That is, by the differences of weather, climate, vegetation, snow cover, etc. between Alpine and non-Alpine climates. Of course, other climatic variables will also account for it to some extent, but no variable will synthetize the difference between non-Alpine and Alpine climate as efficiently as the elevation (by definition !). Therefore, any effect of the elevation in SDM models at this scale will summarize the difference of environments (vegetation, climate, etc.) between Alpine and non-Alpine environments. Therefore, if -- for example -- a species is absent or rarer in the Alps than in the rest of the continent, it will be because the Alpine climate is less suitable for the species than non-Alpine climates (whatever the reason, e.g. too much snow in winter, rocky soils, etc.). This will be synthetized by a negative effect of the elevation in the SDM

On the other hand, the elevation, in the Azores archipelago, has a very different meaning. In these islands, high elevation areas are not characterized by a "more Alpine climate" than other areas. It has a very different meaning. In these islands, for example, there is probably a very strong correlation between elevation and the distance to the sea. So that elevation here rather represents this distance (e.g. further from the sea = less urban, etc. -- maybe a better habitat for the species?). Thus the meaning of elevation for the species is not the same, just because we do not work at the same scale. Of course, my example here describes a fictious species (I do not know what are the selection patterns for weasel and ferret at this scale), but it illustrates why results are expected to vary between two scales, even within the native range (e.g. the results will not be the same if we compare a model fitted on the whole Europe and an model fitted e.g. on similar sized areas in continental Portugal).

Authors: Indeed, interpretation of the effect of elevation is different for the Alps and the Azores, but we do not believe this is linked to scale but rather we are talking about the second order from Johnson (1980) and the local conditions in Soberon’s work or the geographic factors that determine species distributions in mountains from Grinnell or Humboldt. Indeed, we are dealing with mountains in islands and not continental mountains. We carefully interpret the coefficients (direction and magnitude) in light of the context in which they occur. Often the proxy elevation is used to represent the variety of conditions that occur with zonations in mountains (Schrodt et al. 2019). The specific mechanisms by which individuals and species respond to these are seldom not represented in SDMs. This gets back to the use of SDM to predict distribution, habitat selection, or individual movements – and in the ecological literature for each of these processes there are different types of models. SDM, as the name states, refers to geographical distributions and some authors even argue they should not be used to infer habitat use or animal movements as inferred from Johnson (1980) orders of habitat selection across scales.

Schrodt F, Santos MJ, Bailey JJ, Field R. Challenges and opportunities for biogeography—What can we still learn from von Humboldt?. J Biogeogr. 2019; 46(8): 1631−1642.

The same is true for e.g. climatic variables. The effect of these variables on a continental scale will likely reflect the differences in densities between the different climates in Europe. If a species is less abundant in e.g. areas with continental climates (with less precipitation), then the effect of clim_bio12 in a SDM on a continental scale will mainly represent this large scale pattern. At the scale of the Azores Islands, this variable will not have the same meaning, as there is no continental climate in the Azores islands. Etc.

Authors: Indeed, interpretation of the effect of any variable is different due to the context and in the discussion we state how we interpret these effects. This is true for any SDM study.

If the aim is to predict the introduced range from data collected in the native range, it would make much more sense to compare this archipelago with areas *of similar sizes* located in similar climates, with a similar elevation range, etc.

Authors: Thank you for the suggestion. As mentioned above we experimented with it, and our conclusion was that the criteria to select areas of similar extent to that of the Azores, located in similar climates and with a similar elevation range would be extremely arbitrary. First, selecting random 2,250 km2 areas in a continent that is continuous would suffer from many of the issues you mention above as to why we should do it. Second, there are no similar climates to the Azores in continental Europe, and third, as in your explanation, the Alps and the mountains of the Azores are different in the way they affect species distributions therefore not warranting making these choices. Therefore we chose to represent our native area by non-connected cells, i.e., sampling on the native range. 

I suggested this approach in my previous review, but the authors replied:

"We concluded that it would be difficult to justify selecting sub areas within continental Europe that would match the size of the Azores to conduct the analysis as suggested. This is because there could be an even bigger effect due to this selection than the potential effects of modeling a smaller area using information from a larger area."

I do not understand why it would be difficult to justify it. I do not think that there would be a bigger effect in this selection. The authors have a very precise description of their study area according to numerous environmental variables (climate, elevation, etc.). It would be easy to select randomly numerous places of similar sizes in Europe. The standardized environmental variables (i.e. minus the mean and divided by the standard deviation) then each define a dimension in a multidimensional space (one dimension is the average temperature, one is the elevation, on is the precipitation, etc.). Each one of these random areas would define a point in this space. The Azores archipelago also defines a point in this space. Then, we can select a sample of -- say -- 10 random places with the smallest Euclidean distance in this multidimensional space to Azores. The resulting sample will be a random and objective sample of by areas taken in the native range with sizes and environmental conditions similar to the target area. This would be a better approach in my opinion: if the aim is to infer the effect of a factor on a process by comparing two sets of areas -- one with the factor and one without -- it is better to design the study so that only that factor varies and the other variables are identical. Here, to try to find one or several study areas in the native range with sizes and conditions similar to those of the Azores archipelago, with only "introduced/native" differing. On the other hand, as exemplified above, ignoring the effect of scale will lead to erroneous conclusions.

Authors: Thanks for your suggestion. Although we respectfully disagree, we tried to perform the suggested analysis. First, we created a polygon with similar extension to the Azores (2,250 km2; 75x30 cells). After that, we randomly generated 100 polygons across whole Europe (see Fig R2 in "Response to Reviewers" Letter). The polygons were generated in areas with presence data (i.e. all polygons possess occurrence-records of the target species). We calculated the mean value of the bioclimatic variables for each polygon. Means of the bioclimatic variables were also calculated for the Azores. Finally, we performed a PCA analysis and selected the polygons with the smallest Euclidean distance in this multidimensional space to Azores (see Fig R3 in "Response to Reviewers" Letter). 

However, due to the large extension of Europe and the disperse spatial location of the occurrence records, the polygon with most occurrence data had only 30 records (our native area, i.e., Europe, has more than 1,500 occurrence records). 

The results of the PCA showed that the polygons with more similar conditions to the Azores had less than 10 occurrence records (polygons ID 10 and 9, see Fig R3). These polygons were located in the occidental area of the Iberian peninsula, near to the coast line.

In conclusion, although with the analysis suggested by the reviewer the extent of the study was similar to the Azores (2,250 km2 vs 2,300 km2), in our study the extent of the native area used to perform the models was also similar (3,000; i.e.10,000 total background points, from which MaxEnt randomly selected 30%). However, the number of presence data used to run the models is dramatically lower using 75x30 polygons in comparison with our selected native-area.

The authors further note:

"Further, we believe that this suggestion is not in line with other approaches to model invasive species ranges. Several authors have also modeled invasive species range from information retrieved from the native range (e.g., Broennimann et al., 2007; Fitzpatrick et al., 2007; Rödder et al., 2009; Bidinger et al., 2012); This is because invasive species are yet to be in equilibrium with their environment in the invaded ranges thus becoming even more important to include a broader set of parameter ranges from the native range to predict areas with potential for invasion."

However, none of these works compare areas with so large size differences. Broenniman et al. (2007) predict the distribution of spotted knapweed in North America with a model fitted with data collected in its native range in western Europe, two areas covering similar sizes. Fitzpatrick et al. (2007) compare the distribution of red ants in tropical south America and tropical north America, again at similar scales and in similar climates. Rödder et al. (2009) study the slider turtle in their native range (North America), and the invasive range is also on a continental scale. Finally, Bidinger et al. (2012) studies the distribution of Harlequin ladybird in their native range in eastern Asia, and in their introduced range (area of similar size in Europe).

I fully endorse the aim of the present study. From a conservation as well as ecological point of view, it is essential to find a way to predict the introduced range from the native range of invasive species. I agree on the importance of understanding how the introduced range might differ from the native range to identify how a species adapt to a new environment. I do not disagree on the aim, but on the method. The problem of scale is not a minor one in ecology. Comparing two areas of very different sizes amounts to compare a process at two very different scales. Therefore, different results are expected, even if there is no difference between native and introduced ranges.

Authors: At the moment we have conducted three analysis:

1. Analysis with 10,000 cell across Europe and the Azores ─ in the original manuscript;

2. Additional analysis of the occidental part of the Iberian peninsula as the native-area (comprising mainly the north-west of Spain and Portugal) and the Azores. (not in the manuscript). We selected this area because: (1) it comprises a large area to where both species are native, thus including most relevant ecological requirements of the study species; and (2) it is the closest native distribution area for the study species. Now, the results of the PCA (Fig R3) showed that this native area has similar environmental conditions to the Azores. We received criticism for this approach because it is unknown where the original introduced individuals originated from, as they easily could have originated from another region in their large native range. By restricting the study to a small portion of the native range of both species the authors are not capturing the full range of environments these species can inhabit and thus will impact the model predictions and comparisons with the invaded range model;

3. We attempted the proposed approach of generating 2,250 km2 areas and select those with closer climatic properties to the Azores, but as explained above we ran into limitations of sample sizes

So we are facing a trade-off, of restricting analyses to randomly selected areas of the same size of the Azores (as mentioned above, a bit difficult to justify their choice) and have small samples, or run models sampling from the native range in a similar area as the Azores (10,000 cells is a similar number of cells to those used for the Azores models). Therefore, selecting a smaller area, the number of native records would be dramatically lower and would not reflect the native range. This is the reason why other reply was "we believe that this suggestion is not in line with other approaches to model invasive species ranges [...] This is because invasive species are yet to be in equilibrium with their environment in the invaded ranges thus becoming even more important to include a broader set of parameter ranges from the native range to predict areas with potential for invasion."

******

** On the correction of sampling effort.

The authors rightly explained that the biased data collection characterizing the dataset would lead to biased inference if it was not taken into account. They use the "target group" method of Philips et al. (2009) to collect a biased sample of background points exhibiting the same bias as the presence records. The authors did not describe clearly their approach in the previous version of the manuscript. It is now more clearly described.

The "target group" approach of Philips et al. aims at distinguishing whether the absence of detection of the focus species at one place is caused by the absence of the species itself or by the absence of data collection. The idea is to define a "target group" containing many species for which we think that the collection or observation activities of collectors is similar to those of the focus species. For example, to model the SDM of a particular bird species in a citizen science program, it would make sense to use the whole set of bird species studied in the citizen science program as the target group used to select background points. The hope is that, at any point where data collection for the focus species has occurred, at least one species of the target group was present and reported by the same data collection (even if the focus species itself is not). So that the presence of a species of the target group and the absence of the focus species ensures that the absence of reported detection for this focus species is likely due to the actual absence of the species.

This method allows to correct -- to some extent, it is impossible to completely account for all the bias in such contexts -- the collection bias because the target group is usually made of a large number of species, and at least one is supposed to be present where data collection has occurred. In the paper of Philips, for example, the target group contains from 7 to 52 species.

In the present paper, the target group was defined by just the ferret and the weasel. This is a small target group! It means that any area unsuitable for both species, but where data collection has actually occurred will not be present in the data. The background sample will be defined by the set of habitat conditions allowing the presence of the weasel and/or the ferret (well, a biased sample of it, but this is the aim of the target group method to obtain such a biased sample of background points). So that modelling the presence of the weasel (resp. ferret) in a set of locations defined so that either the ferret or the weasel are present, is not a model of the species distribution. It is a model of the niche difference between the two species: the model predicts the probability of presence of one species given that at least one of the two species is present.

So that the "native" model describes the niche differences between ferret and weasel modelled at the scale of the continent species, whereas the "introduced" model describes theses differences at the scale of the archipelago. In other words, the models do not focus on the potential distribution of the species as indicated in the paper, but on the differences between the two species. Which may be of interest, but is not the actual aim of the study.

Authors: We did not use exactly the target-group method proposed by Phillips et al. 2009. We only based our biased background data on the reasoning of Phillips et al. 2009 about the necessity to account for bias in the selection of the background points according to presence data, but we did not considered a target-group species to select presence-absence data. We used the MaxEnt software to perform the models, which takes as input presence-only data and a set of environmental variables across a user-defined area, in our case, 10,000 isolated background points across whole Europe. 

We clarified this in the new version of the ESM1 File. Now it reads: "Phillips et al. 2009, proposed to select background data that exhibits the same bias as the presence data. For example, if the presence data are taken from a determined portion of the study area, then the background data should be taken from the same areas (Ferrier et al., 2002; Phillips et al., 2009). Following this reasoning, we (i) created a grid comprising all native area (i.e., Europe), (ii) randomly selected 10,000 cells within <10 km of the species presence cells, and (iii) used 3,000 random background points from this selection to run the native-based models in MaxEnt."

Phillips SJ, Dudík M, Elith J, Graham CH, Lehmann A, Leathwick J, Ferrier S. Sample selection bias and presence‐only distribution models: implications for background and pseudo‐absence data. Ecol Appl. 2009; 19(1): 181−197.

Ferrier S, Watson G, Pearce J, Drielsma M. Extended statistical approaches to modelling spatial pattern in biodiversity in northeast New South Wales. I. Species-level modelling. Biodivers Conserv. 2002; 11(12): 2275−2307.

******

** References

Dungan, J.; Perry, J.; Dale, M.; Legendre, P.; Citron-Pousty, S.; Fortin, M.; Jakomulska, A.; Miriti, M. & Rosenberg, M. 2002. A balanced view of scale in spatial statistical analysis. Ecography 25, 626-640

Johnson, D. 1980. The comparison of usage and availability measurements for evaluating resource preference. Ecology 61, 65-71

Levin, S. 1992. The problem of pattern and scale in Ecology. Ecology 73, 1943-1967

Soberon, J. & Peterson, A. 2005. Interpretation of models of fundamental ecological niches and species' distributional areas. Biodiversity Informatics 2, 1-10

Pearce, J. & Boyce, M. 2006. Modelling distribution and abundance with presence-only data. Journal of Applied Ecology 43, 405-412

Elith, J. & Leathwick, J. R. 2009. Species distribution models: ecological explanation and prediction across space and time. Annual Review of Ecology, Evolution, and Systematics 40, 677

 

Reviewer #2:

The authors have addressed the concerns expressed in the previous review. The explanation and map provided in Appendix S1 make the procedure much clearer and more transparent, and the dataset provided in appendix S2 adds significant value and usefulness to the manuscript. All other supplementary materials also help make the methodology clearer. Although not everything was done the way I would have done it myself, I am generally satisfied with the current version of the manuscript and I believe the authors have appropriately explained and defended their work.

Authors: Thanks for your comment. We are pleased that you are satisfied with the manuscript.

---

## [Editor Report · Decision Letter 2]

23 Jul 2020

Modelling the distribution of Mustela nivalis and M. putorius in the Azores archipelago based on native and introduced ranges

PONE-D-19-33731R2

Dear Dr. Lamelas-López,

We’re pleased to inform you that your manuscript has been judged scientifically suitable for publication and will be formally accepted for publication once it meets all outstanding technical requirements.

Kind regards,

Ulrike Gertrud Munderloh, Ph.D.

Academic Editor

PLOS ONE
---

## [Editor Report · Acceptance letter]

27 Jul 2020

PONE-D-19-33731R2 

Modelling the distribution of Mustela nivalis and M. putorius in the Azores archipelago based on native and introduced ranges 

Dear Dr. Lamelas-López:

I'm pleased to inform you that your manuscript has been deemed suitable for publication in PLOS ONE. Congratulations! Your manuscript is now with our production department. 

Kind regards, 

on behalf of

Dr. Ulrike Gertrud Munderloh 

Academic Editor

PLOS ONE